# How to not induce SNAs: The insufficiency of directional force

A. Michirev[1][☉], K. Kühne[2][☉]*, O. Lindemann[3], M. H. Fischer[2][‡], M. Raab[1,4][‡]

**1** Department of Performance Psychology, German Sport University Cologne, Cologne, Germany, **2** Division of Cognitive Sciences, University of Potsdam, Potsdam, Germany, **3** Department of Psychology, Education and Child Studies Erasmus University, Rotterdam, Netherlands, **4** School of Applied Sciences, London South Bank University, London, United Kingdom

☉ These authors contributed equally to this work.
‡ These authors also contributed equally to this work.
* kkuehne@uni-potsdam.de

## Abstract

People respond faster to smaller numbers in their left space and to larger numbers in their right space. Here we argue that movements in space contribute to the formation of spatial-numerical associations (SNAs). We studied the impact of continuous isometric forces along the horizontal or vertical cardinal axes on SNAs while participants performed random number production and arithmetic verification tasks. Our results suggest that such isometric directional force do not suffice to induce SNAs.

## Introduction

Cognitive representations of space, time, and number provide a mental structure for how we humans experience our environment [1]. One of those structures is the Spatial-Numerical Associations of Response Codes (SNARC, [2]) effect that describes a representation of magnitude meaning along a horizontal mental number line with smaller numbers represented to the left of larger numbers. Experiments testing the SNARC effect typically rely on bi-manual responses or spatial cueing in a reaction time paradigm across different tasks (e.g., speeded parity judgments, magnitude classifications and simple mental arithmetic; for reviews see [3,4]). The reaction time paradigm is widely accepted and provides evidence for the association of smaller numbers with left space and larger numbers with right space [4].

Originally, the SNARC effect was attributed to reading and writing directions resulting from cultural exposure [2]. However, subsequent research on SNAs produced several alternative hypotheses that consider biological, cognitive, behavioural, and embodied cognition approaches. For example, a biological approach utilizes hemispheric asymmetry of the brain [5–7] whereby the right and left hemispheres are tuned for processing small and large numerosities, respectively. Therefore, the natural spatial scanning direction would go from left to right, which associates left with small and right with large numerosities.

A cognitive approach considers positional coding of numbers in serial working memory and utilizes an experienced-based spatial template [8]. This spatial template is then used to arrange information according to the situation and relevant references. For instance, when the

**Data Availability Statement:** All the data are uploaded on OSF (https://osf.io/v7dyj/).

**Funding:** This research received by the German Research Foundation (DFG) grant RA 940/16-2 awarded to MR and grant FI 1915/5-2 to MHF. The

funders had no role in study design, data collection and analysis, decision to publish, or preparation of the manuscript.

**Competing interests:** The authors have declared that no competing interests exist.

number range is between 1 and 9, then the number 5 would provide the reference. In this scenario numbers 1–4 will be placed on the left of the 5 while numbers 6–9 to the right. Together, reading and writing direction, hemispheric asymmetry, and the serial working memory approaches are capable of explaining SNAs along the cardinal horizontal axis. Additionally, a behavioural approach considers stimulus-response associations with polarity correspondence in binary tasks [9,10]. Hereby a stimulus exists in a dimension of polar opposites, such as a number being either small (negative pole) or large (positive pole). In a SNARC paradigm, a large number is paired with the response to the right (i.e., both represent positive poles). Therefore, when the polarities are matched they facilitate the response that can explain the SNARC effect. Note that, within the polarity correspondence account, the SNARC is produced by stimulus-response correspondence and not due to an association of space and numbers. Therefore, SNARC would not be limited to the horizontal axis (but see [11] for empirical evidence against the polarity correspondence account).

Linguistic metaphors such as "more is up" are widely found and seem to reflect universal physical laws that explain SNAs on the vertical axis [12,13]. The embodied cognition approach encompasses such linguistic practice and suggests that all concepts, and therefore also SNAs, are bi-directionally related to sensory and motor experiences so that specific activation in one domain (experiential or conceptual) translates to the other [12,13]. Applied to number symbols, the role of the body, especially the systematic use of fingers during number learning, counting [14] and gesturing (e.g., [15]), seems to contribute to the development of numerical cognition (for an overview see [16,17]). The use of fingers seems to be deeply rooted in using the hands to count, which was already evident during Palaeolithic times [18]. In this line of reasoning, the finger-counting hypothesis [14,19] predicts the SNARC effect along the horizontal axes because in Western countries people usually start counting on their left hand and with the left thumb (left-starters), therefore associating small numbers with their left space. Finger-counting then progresses from left to right, just as the typical left-to-right SNARC. Vertical associations instead reflect other sensory and motor experiences, such as the growth of piles during object accumulation or the rise of water in a container, which inspired the linguistic metaphors mentioned above. Notably, there is an asymmetrical relationship between space and numbers, in which the understanding of space is more fundamental. The meaning of numbers is often based on the experiences of space while this is not necessarily the case vice versa [12,13]. Together, the embodied cognition approach predicts SNAs on the horizontal and the vertical axes based on different and independent mechanisms.

Recently, more evidence accumulated towards the situatedness and flexibility of finger counting and the starting hand: while finger counting habits seem to be rather stable over time [20] they are also flexible depending on the situation [20,21]. Moreover, even in Western individuals, for example, Hungarians, Germans and Italians, the right hand [22,23] or either hand [24] is used to start counting while in Middle Eastern countries people tend to start counting on their right hand (with their small finger, see [25]). Overall, while the evidence on the direction of counting (starting left or right) is rather mixed and emphasizes situatedness [26], the unequivocal involvement of fingers in SNAs point towards the embodied nature of numerical cognition.

Intriguingly, the SNARC effect was recently shown to exist in 3-dimensional space [27] for each of the three cardinal axes (horizontal, vertical and sagittal) whereby left/down/near are associated with smaller and right/up/far with larger numbers (for review see [28]). The authors [27] have interpreted the 3-dimensional SNARC as indicating the existence of three independent mental number lines. These findings are of particular interest because they provide counter-evidence for all the above accounts. For instance, extending the horizontal mental number line to 3-dimensional space opposes the accounts of reading and writing directions

[2], hemispheric asymmetry [5–7] and positional coding of numbers in serial working memory [8] because those accounts only explain SNAs along the horizontal cardinal axis. Furthermore, the independence of the three hypothesized mental number lines opposes the polarity correspondence account [9,10] that assumes symmetry of all cardinal axes. Additionally, while the embodied cognition approach can make predictions of SNAs on the horizontal and the vertical axes, it lacks a strong theoretical reason to predict SNAs on the sagittal axis [29].

The goal of the present study is to test the boundary conditions under which the association of numbers and space is manifested in behaviour. We adopted the embodied cognition approach and focused on the horizontal and vertical axes. Examining prototypical experimental conditions under which SNAs are found and not found can help identify a common ingredient across existing experiments (for recent null results for horizontal SNAs see [30,31]). By this rationale, the common ingredient for most studies seems to be the activation of spatial information, namely under the experimental conditions of response space (lateral responses), presentation space (lateralized stimuli), or spatial cueing (arrows and/or instructions) during stimulus presentation (see Fig 1 for a visual overview of prototypical laboratory set-ups).

Response space seems to contribute to a range of experiments using bimanual responses (see Fig 1, examples 1 to 4). For example, the original horizontal SNARC [2] was observed during the central presentation of numbers when participants responded bimanually by pressing keys arranged laterally on the left and right side. Additionally, a horizontal SNARC emerges

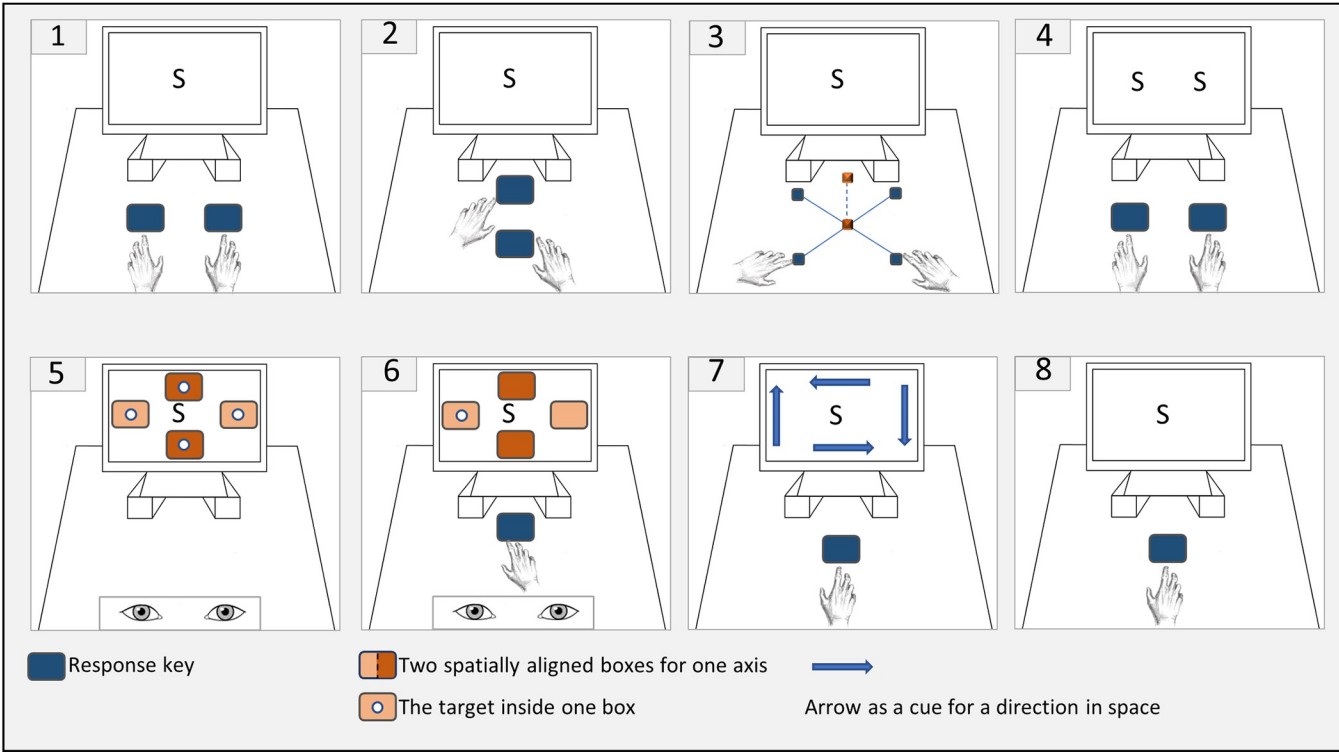

**Fig 1. Prototypical experimental designs reporting SNAs.** 1 Lateralized response space with centralized stimuli presentations on the horizontal axis [2]. 2 Lateralized response space with centralized stimuli presentations. The authors wanted to access the vertical axis, however, the spatial alignment of response keys depicts the sagittal axis (e.g., [32–34]). 3 Lateralized response space with centralized stimuli presentations along all three cardinal axes [27]. 4 Lateralized response space with lateralized stimuli presentations (e.g., [35]). 5 Lateralized response space with centralized stimuli presentations on the horizontal or the vertical axis (saccadic movement to the target; e.g., [36,37]). 6 Centralized response space with lateralized target presentations on the horizontal or the vertical axis (visual target detection by key response; e.g., [35,38]). 7 Centralized response space with centralized stimuli presentations but spatial cueing [31]. 8 Centralized response space with centralized stimuli presentations where no SNAs were found [30].

when bimanual response space, as above, is paired with lateralized stimuli presentation (simultaneously displayed digit pairs for magnitude comparison, [35]). Furthermore, response space is not exclusive for the horizontal axis and is present in SNARC research with the vertical axis. However, as a keyboard is commonly used to record responses, this is particularly questionable because the up and down arrows on the keyboard operate the cursor vertically on the computer screen but are located along the sagittal axis (see Fig 1, example 2). In order to have a purer assessment of SNAs along the vertical axis, eye tracking experiments might be more suited. However, even under these experimental conditions spatially aligned targets are typically needed (but see experiments using auditory stimuli [39,40]). For example, when participants are presented with centrally displayed stimuli they are required to respond by saccadic movements to the target that appears inside spatially aligned boxes to either left and right or up and down (see Fig 1, example 5). Finally, response space is also presented in the 3-dimensional SNARC as participants used bimanual responses along the three cardinal axes (see Fig 1, example 3).

Very recently, a new study challenged the importance of active responses in the response space. Miklashevsky et al. [41] relied on a paradigm of passive and continuous isometric force readings during numerical tasks. Participants produced more force in the left hand while processing smaller numbers and more force in the right hand while processing larger numbers. These results suggest that active responses might not be required to induce SNAs. However, the authors did not control for spatial effects in their design. Specifically, in the response space, the authors utilized two sensors that were held in the left and in the right hand, thus being laterally displaced. Therefore, spatial information in the response space was apparently sufficient to elicit SNAs in their study.

Alternatively to the response space, the presentation space in which magnitudes are displayed can create SNAs. For instance, whenever experiments have centralized their response space in a reaction time paradigm (e.g., by using the space bar) they instead included laterally presented targets (see Fig 1, example 6; target detection paradigm). Such designs remove the response codes; however, they introduce spatial codes in the targets. For example, a stimulus is presented on the screen together with laterally aligned boxes (either left/right or down/up). Then a target appears inside one of these boxes (spatial code) that participants respond to by pressing the space bar (e.g.,[42]).

Recently, the target detection paradigm was extended to test the temporal progression of number processing during mental arithmetic (e.g., [38]). During such temporal progression a systematic bias of accepting larger values in addition and smaller values in subtraction problems was observed (Operational Momentum, see [43]) and interpreted as a spatial shift of attention on the mental number line that relies on the spatial nature of number representations. However, the onset time of these SNAs is currently debated in the literature (e.g., [38]). For example, several studies report SNAs after the appearance of single digits [2,42,44]. Others report these SNAs only after both the first operand and the operator of an arithmetic problem had been presented, indicating that spatial associations emerge with arithmetic operators (Operation Sign Space Association; [38,45,46]). Yet another study [47] reported spatially biased eye behaviour only after all problem elements were known to their participants. Therefore, the target detection paradigm provides a useful tool that can help track SNAs on a temporal continuum. However, this paradigm also suffers from spatial confounds as the target is displayed laterally and therefore carries spatial information across either the horizontal or vertical axis.

Until now we have presented prototypical examples of experimental conditions under which SNAs emerged, considering both presentation and response space. Next, we consider the SNAs under experimental conditions that have controlled for response and presentation

space. These experiments removed spatial information from their SNA assessment by showing non-lateralized numbers and avoiding lateralized responses. If our rationale is correct that spatial information contributes to SNAs, then SNAs should not emerge under such spatially unbiased assessments.

Indeed, Pinto et al. [30] (see Fig 1, example 8) deployed such a "purified" assessment of the SNAs under conditions of central presentation space but with varied response space: one task relied on lateralized responses and the second task relied on central responses. The authors showed that only the task that required bi-manual responses (lateralized response space) produced the typical left-to-right SNARC. The other task that aligned response space with the presentation space did not find SNAs ("only press the centralized space bar if the number is larger/smaller than 5, otherwise do not react"). The authors' interpretation was that the numbers will only spatially align across the mental number line when both, number magnitude is activated and spatial response codes are present (for experimental designs that also centralized the presentation and response space see [48,49]).

Another study ([31], see Fig 1, example 7) centralized both presentation and response space. However, this study used spatial cueing with arrows pointing to the left, right, down or up. Despite this spatial cueing the authors found no horizontal SNAs in this version of the parity judgment tasks. Consequently, these findings suggest that the much-studied horizontal SNAs might be ephemeral artefacts of spatial task demands. Together, the above-presented studies point to the conclusion that both, number magnitude and spatial information must be coactivated to produce reliable SNAs [30,31,48,49]. Intriguingly, in the magnitude judgment tasks, the up-to-down cueing succeeded and vertical SNAs persisted and were always stronger than horizontal SNAs.

One possible explanation for why vertical SNAs remained stronger than horizontal SNAs could be the hierarchical nature of body-related knowledge representations involved in the task, according to which different mechanisms induce different types of SNAs, referred to as grounded, embodied, or situated cognition [29]. For example, the finger-counting hypothesis considers *embodied* sensorimotor experiences and cultural influences [14,19]: The hand that is used to start to count defines (or strongly suggests) the direction of horizontal SNAs. Instead, vertical SNAs are *grounded* in deeply rooted magnitude-space associations that reflect universal physical laws and inspired ubiquitous linguistic metaphors, such as "more is up" [12,13]. Indeed, expressions such as "high number" or "low number" reflect vertical SNAs while no such linguistic connection exists for horizontal SNAs [28]. These linguistic practices that describe vertical SNAs differ in their contribution to the embodiment of numbers compared to the horizontal SNAs that seem to rely more on sensorimotor experiences such as finger counting. Moreover, situated factors like interoceptive signals play an additional role in the perception and production of numbers that also might contribute to SNAs ([50]; for review see [51]). Therefore, vertical SNAs might be more general or stable than horizontal SNAs, which could in turn explain why spatial cueing was enough to elicit vertical SNAs but not horizontal SNAs under the exclusion of spatial responses.

Overall, the embodiment perspective and its bi-directionality (e.g., [29,52,53]) can shed new light on the association between space and numbers. So far, we have discussed one of the two directions of concept-motor interactions, namely how activations of magnitude representations can affect motor responses and produce spatial biases. The second direction is reversed and predicts how movements can, in turn, affect conceptual representations and produce number magnitude biases. The bi-directional link is part of A Theory of Magnitude (ATOM, see [54,55]) that proposes a generalized magnitude representation system across perception, cognition and action.

To test the bi-directionality assumption of the concept-motor interactions, Random Number Generation (RNG) can be utilized. Hereby, the RNG task assesses the cognitive availability of magnitude-related concepts of participants who generate and produce numbers. Specifically, participants are required to produce numbers in specific ranges as randomly as possible [56,57]. RNG can test the concept-motor interactions because number magnitude can produce spatial biases in movement, while movements in space can, in turn, produce magnitude biases. For instance, on the one hand, during a walking task random number generation affects the decision to take either a left or right turn [58], thus producing spatial biases in movement. This is in line with the SNAs research explaining how magnitude activation (small/large numbers) affects movement in space (left/right turns). On the other hand, deciding to take a left/right turn also affects RNG [58]. Other movements through space generally seem to affect RNG as well. For instance, healthy adults generate larger numbers after a right/up head turn and smaller numbers after a left/down head turn ([59]; for RNG across the horizontal axis see also [56]; for review see [28]). Interestingly, such RNG patterns also transfer from active to passive movements in space. For example, passively moving the participants through space produces magnitude biases along the horizontal and vertical axis, however, not along the sagittal axis (passive whole-body movements, see [60]).

While the RNG task is well-suited for testing concept-motor interactions, we note that none of the above-presented RNG studies controlled for spatial influences. Therefore, the RNG task is another spatial task producing or relying on movements through space such as a rhythmic head movement to the left/right (e.g., [56]) or a simple button press in a reaction time paradigm representing a location-related action (see Fig 1, examples 1 to 4).

Generally, spatial information in the design (e.g., movements through space or responses in space) seems to be essential when investigating SNAs, both as a response producing SNAs and to activate SNAs. One line of the above-described paradigms utilizes button presses, lateral eye movements, or movements that include location-related actions measured by the task. The other line of paradigms rather produces movements in response to number generation. Again, the common ingredient seems to be movement in space. Therefore, it might be movements in space that activate the spatial nature of numbers, thus enabling SNAs.

Considering all the presented evidence, we can conclude that spatial information is omni-present in many experimental paradigms that study SNAs. First, spatial information in the response space plays a major role in accessing SNAs along all three cardinal axes [27]. Second, movement through space can activate SNAs with the RNG task across the horizontal and vertical axes [56,59] but not on the sagittal axes [60]. Third, cognitive spatial cueing induces stronger vertical than horizontal SNAs during centralized stimulus presentation and responses [31]. Fourth, taking a turn to the left/right affects RNG [58]. Based on this we follow the logic of the falsification by elimination approach [61] in which we eliminate spatial task demands from the present experimental design and ask the following question: Will directional but non-spatial movement along the horizontal and vertical axes, thus <u>pressing into a direction of left, right, up, or down</u>, elicit SNAs?

In the present experiment we centralized the presentation and the response space. We asked participants to press against a sensor surface, thus recording their isometric force production. Such isometric force was exerted constantly and continuously on the sensor surface in a direction on the horizontal (left or right) and the vertical (up or down) axes during two tasks. Hereby, we utilized the same force sensors as Miklashevsky et al. [41] but altered the paradigm to record the production of isometric force by one-handed presses into a direction instead of grip force (from here on described as "isometric directional force"). Therefore, our isometric force paradigm does not require the execution of a lateralized response after processing a number. Instead, isometric force provides a continuous and implicit measurement of

spatial activation during number processing. First, with the RNG task we aimed to test if isometric force, exerted into a direction along the horizontal (left or right) or the vertical (up or down) axis, was sufficient to produce SNAs. Second, we aimed to test if RNG itself can affect isometric force production. This approach allows to test the bi-directionality of the concept-motor interactions. The following hypotheses were not preregistered but formulated a priori and derived from theory and previous empirical work.

H1. We hypothesize that isometric directional force activates spatial codes that affect the production of random numbers, thus testing how isometric directional force affects the conceptual representation of numbers. That is, participants are expected to produce more small numbers when pressing in the direction left/down and more large numbers when pressing in the direction right/up [56,59].

H2. We hypothesize that generating numbers by itself affects motor activation [58], thus testing how conceptual representation of numbers affects force production. Generating numbers is assumed to activate within-magnitude associations (ATOM, [54,55]), which in turn, are hypothesized to modulate the amplitude of applied isometric force. Hereby, generating smaller numbers should produce smaller force while generating larger numbers should produce larger force when simultaneously pressing against the surface of a force sensor [54,55].

As an additional research question of the experiment, we aimed to explore SNAs on a temporal continuum during mental arithmetic, thus testing how activation of number concepts affect force production during mental arithmetic. The rationale is similar to that of the target detection paradigm (see Fig 1, example 6). However, we removed spatial information from the presentation and the response space. Therefore, instead of providing spatial information in the form of targets to probe these SNAs, we again relied on central stimuli presentation and isometric directional force. In this task, spatial associations of numbers could become activated in relation to the arithmetic procedure; namely either during the first operand, or the operation sign, or the second operand, or during the answer. Similarly to the reasoning in H2, number meaning related to the task should be expressed by the participants' spontaneous force production, with smaller numbers generating smaller force and large numbers generating larger force [54,55].

Taken together, this study aims to test SNAs across the horizontal and the vertical axes by applying isometric directional force. This design allows to exclude spatial task demands in the responses by excluding movements through space and in the stimuli by excluding spatially placed targets. Hereby, the RNG task will answer the question if isometric directional force suffices to elicit SNAs while the arithmetic task will help with tracking the dynamics of SNAs across the arithmetical task.

## General method

### Participants

The total sample size is N = 72 (41 female, mean age 23, median 23, range 17–35 years). All participants were native German speakers and right-handed [the German version of the Lateral Preference Inventory [62] showed a mean of 82,3 and a median of 86.7. The sample was obtained across two cooperating but independent laboratories to ensure the generalizability and replicability of findings across both laboratories and experimental settings. Hereby, Laboratory 1 was allocated at the German Sport University Cologne and Laboratory 2 at the University of Potsdam. Regarding the power analysis, we applied a conservative effect size estimate of dz = .4 for a two-tailed paired t-test (small vs. large number/compatible and incompatible condition) with power .80 and α-level .05 [63], resulting in a required sample size of 52 for each experiment in each laboratory. This is also a general recommendation for sample size

estimation in a t-test repeated-measures design [64]. After reaching the sample size of 54 (considering possible drop-out) in Laboratory 1, data collection in Laboratory 2 was also stopped for efficiency reasons since the data fit rather the null hypothesis model. In case of a null effect, for repeated-measures designs a group of at least 60 participants is considered to be sufficient for a Bayesian analysis [64].

## Apparatus

A customized home-made wooden/metal box was built to enable isometric directional force (i.e., participants' exerting isometric forces on a fixed force sensor surface) along the vertical and the horizontal axes (two levels each: left/right on a horizontal axis or up/down on the vertical axis). Inside the box a force sensor in the shape of a metallic disc 1.8 cm thick and 4 cm in diameter (FTD-MINI40-E-1.8-M2 from SCHUNK GmbH & Co. KG) was attached to a wall congruent with the experimental force direction condition (see Fig 2, panel 2). Participants placed their right hand inside the box and applied force to the sensor (see Fig 2 for direction left), thus producing isometric directional force measured by the Fz axis of the sensor. With this method, the Fz axis provides most accurate parameters of the force data [65]. The box was adjustable on the vertical and horizontal axes to account for individual hand sizes.

Force measurements were obtained according to the procedure recommended by Nazir et al. [66]. Two computers were used to conduct the experiment: the first presented stimuli, the second recorded the data. The first computer sent an initial trigger through the parallel port and a terminal trigger to the second computer, then the force signal was recorded using Expyriment software [67] at a sampling rate of 1 kHz through an analog-to-digital converter card. Forces were measured in millinewtons (mN).

## Experimental tasks

### 1. Random Number Generation (RNG) task

**1.1 Design.** In order to test the concept-motor interactions of H1 and H2 with the same task, we have utilized two within-subject designs. Both designs utilized one within-subject factor being the force direction (two levels each: left/right on a horizontal axis or up/down on the vertical axis). To test H1, number magnitude constituted the dependent variable with two magnitude levels: small and large. To test H2, number magnitude constituted the independent variable with two magnitude levels: small and large while continuous isometric force was the dependent variable.

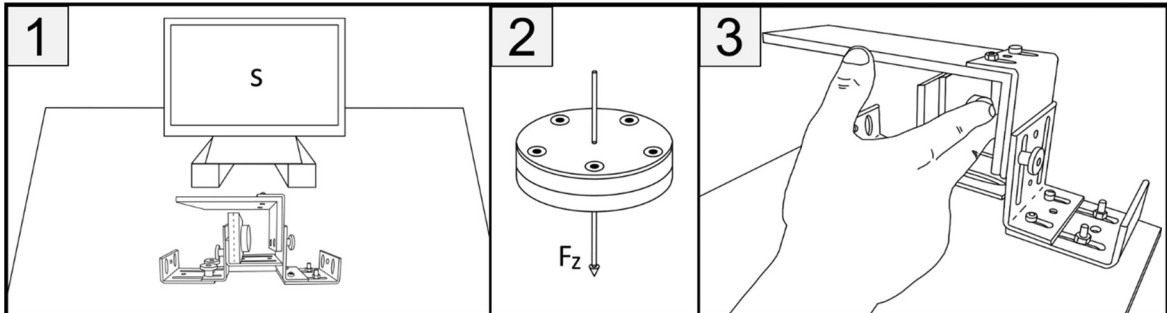

**Fig 2. Conceptual drawing of the experimental set-up.** Panel 1 shows the general experimental set-up with stimuli (S) presented centrally. The box was centralized relative to the stimulus presentation space, allowing the sensor to be placed into one of the four experimental conditions. Panel 2 shows the Fz axis of the sensor that was used to analyse the force data. Panel 3 depicts the left direction with the sensor attached to the left wall.

Additionally to the force directions, there was a control (no-force) condition as a baseline. During the baseline measurement participants were asked to take a relaxed position and rest their hands on their knees to assess unbiased performance. Only the produced numbers were collected and analysed in the baseline condition of H1.

**1.2 Procedure.** Participants were asked to verbally produce a random number from 1 to 9 every 2 seconds. In total participants produced 90 random numbers (cf. [68]) with their eyes closed (for this procedure see as well [50,57]). Closing eyes eliminates possible spatial confounds but maintains isometric directional force as the remaining variable of interest. A metaphor of "drawing numbers from a hat" [50,57,69] was used to instruct participants. A metronome sound, presented with OpenSesame software [70], with a rate of 0.5 Hz was used to indicate the signal for number production every two seconds. Participants responses were recorded with a voice key and noted manually by the experimenter.

**1.3 Data analyses of continuous isometric force.** The following method was not preregistered but closely followed the recommendations of Nazir et al. [66] (for details see S3 File). The selection of time-windows for the force data analysis in the RNG task was based on the trial onset, i.e. the start of the auditory signal in the RNG task after which participants had 2 seconds to state a number. From these 2 seconds we selected the first 700 ms of each trial for calculation because speech production can motorically interfere with manual force [66]. Therefore, we focused our analysis on time-windows prior to speech production, which began on average 576 ms after stimulus onset. We have utilized a cluster permutation analysis (with 5000 permutations and a correction for multiple comparisons) to identify the time-windows of interest during which number magnitude (in operand 1, operand 2, and the result) and the operation (plus, minus) affects force magnitude ([71]; for details see S3 File). In the case of uninformative results of the cluster permutation analysis, five time-windows for each directional condition were selected manually (in ms: 50–150, 150–250, 250–350, 350–450, 450–576). We split the force data into equal chunks of arbitrary length of 100 ms starting from 50 ms to the end of the trial (last time-window being 126 ms long). These time-windows were constructed to capture mainly the effects in the time-windows of 100–250 ms that are associated with semantic activation after a critical stimulus [72].

## 2. Single Digits Arithmetic (SDA) task

**2.1 Design.** A within-subject design with three within-subject factors was used for the SDA task, namely: number magnitude (small vs. large), operation (plus/minus), force direction (two levels: left/right on a horizontal axis or up/down on the vertical axis) with isometric force production as the dependent variable (for overview see S1 Table).

**2.2 Procedure.** Participants were asked to watch mathematical equations (from here on, referred to as trials) in which individual stimuli (operand 1, operation sign, operand 2, equal sign, result) were presented sequentially after each other centrally on the computer screen. Some of the proposed results were wrong. In case of a wrong result participant had to reject the result by saying "No". Each stimulus was shown for 500 ms, then the next stimulus appeared. There was an inter-trial interval of 1500 ms between the trials (see Fig 3).

The task incorporated the Go/No-Go paradigm with wrong mathematical equations (catch trials) to ensure the depth of numerical processing. In GO trials (incorrect result) participants were requested to verbally respond with a "No". In total, the task contained 116 trials (58 unique trials, presented twice) of which 96 were true-trials (correct results/No-Go condition) and 20 were catch-trials (incorrect results/Go condition). The trials were designed by keeping the proposed result in the single digits and positive numbers, therefore limiting the quantity of unique combinations. These 116 trials were presented in random order for each participant

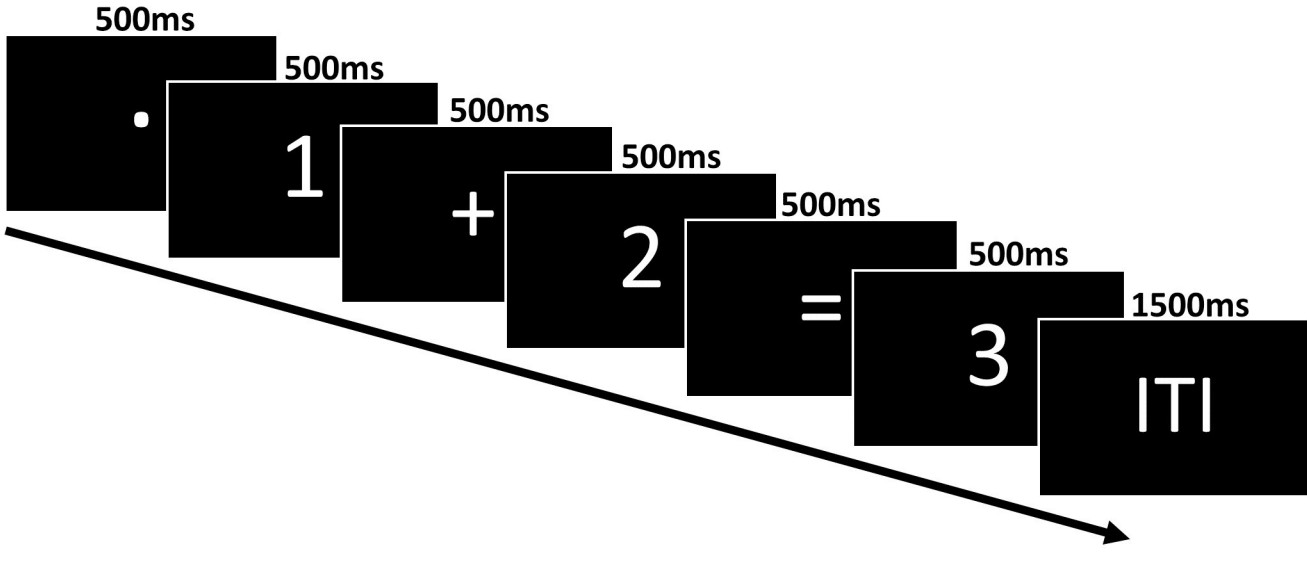

**Fig 3. Time-course of one trial of the SDA task.**

across four blocks, with mandatory self-paced breaks to counter physical fatigue and re-calibrate the force sensor [66]. In total, each participant went through four SDA tasks, one for each force direction (464 total trials per participant). The trials were constructed with single digit operands excluding the number 5 for addition and subtraction (see S2 Table for all stimuli). The digits (size: 100x100 pixel, Calibri font, visual angle of 2.2˚) were created with Adobe Photoshop. They were presented one at time at the centre of the PC screen in white on a black background (in OpenSesame software [70]).

**2.3 Data analysis of continuous isometric force.** In the SDA task, the time-windows of interest were the first 500 ms after each stimulus onset. Identically to the RNG task, the time-window selection was based on either a successful cluster permutation analysis (for details see S3 File) or a manual construction of time-windows (in ms: 50–150, 150–250, 250–350, 350–450 and 400–500) based on the same rationale as the construction of the RNG time-windows. Therefore, for each analysis of operand 1, operator, operand 2 or the answer, we only analysed the 500 ms after that stimulus' onset, respectively. The total time-windows we extracted were the following: operand 1 (0–500 ms), operator (0–1000 ms), operand 2 (0–1500 ms) and answer (0–2500 + 500 ms for the inter-trial interval). This procedure was repeated across all four force-direction conditions.

**2.4. General procedure.** The sequence of force direction conditions and order of tasks was counterbalanced between participants with a Latin square. The experimenter welcomed the participant and seated them in a comfortable chair, at a desk, about 90 cm away from the PC monitor. Participants were informed about the procedure and were asked to sign the informed consent and effector dominance assessment [62].

The experimenter explained the force measurement, as well as the SDA and RNG tasks, using PowerPoint slides supported by a standardized experimental script clarifying any remaining questions if needed. The explanations also covered the exclusion of number 5 and the largest number to be expected to be a 9 (smallest to be 1). Dependent on the force direction condition the participant placed their hand inside the Apparatus. The participant was then instructed to sit as still as possible and to continuously press the force sensor with their index and middle fingers (see Fig 3). The participant then underwent a task-specific familiarization,

including force calibration (See S2 File). Importantly, frequent and self-paced breaks were utilized to minimize participants fatigue levels (recommended by [66]).

Next, participants took a short break and started the experiment with the predefined force direction condition and task. For example, first, the participant was tested in the SDA task in the right force direction condition. Then, upon completion of the SDA task, the participant proceeded with the RNG task in the same force direction condition. Upon completing one force direction condition the experimenter adjusted the Apparatus to the new force direction condition. In total, each participant completed four force direction conditions. The experiment ended after the completion of all four force direction conditions for both the RNG and SDA tasks. Upon completing the session, the participants filled in the demographics form and were rewarded and debriefed. The typical experiment duration was 90 minutes. The study was reviewed and approved by the Ethics Committee of the University of Potsdam, Germany (approval number 21/2019). The participants provided their informed consent to participate in this study.

**2.5 Forms and questionnaires: Demographic form & lateral preference form.** The demographic form recorded the age, gender, native language, whether participants have normal or corrected-to-normal vision, or a history of neurological diseases. We also recorded the use of medications that could affect performance. To assess hand dominance, we have relied on the German version of the Lateral Preference Inventory to ensure that all participants are right-handed [62].

**2.6 Pre-processing force data.** Pre-processing of force data followed the general guidelines proposed by Nazir et al. [66] and is documented in the S3 File. Importantly, due to pre-processing procedures of relevant stimuli in the SDA task (operand 1, the operator, operand 2, and the answer) there was an increased attrition rate for each analysis while progressing through the duration of a trial. Moreover, the attrition rate was not symmetrical across the RNG and the SDA tasks. The accuracy cut-off in the SDA task was set to 90% and was applied for all four directional force conditions separately. In RNG task there was no accuracy cut-off, however two participants were excluded because they failed to comply with the task (they generated numbers outside the range or counted systematically up or down).

The experimental protocols of both cooperating but independent laboratories were identical, however, due to possible soft and hardware differences the force data from both laboratories was not collapsed into one data set. This was done to ensure that measuring reliability regarding temporal recordings remains across the individual data sets. Instead, identical analyses were performed on both data sets. Then, the Bayes factors from both analyses were multiplied by each other. This procedure allowed to use Bayes factors as inclusion of prior knowledge [73]. Therefore, the main text includes the combined Bayes tables while the S1 File shows individual analyses from both laboratories. The default width parameter of .707 was used in all analyses since we did not have strong assumptions about the distribution. Moreover, it was shown that the prior width has little impact on the Bayes factor [74]. All analyses were done within each independent axis (horizontal or vertical).

## Results

### 1. RNG

**1.1 Numbers.** To test SNAs induction in terms of generated numbers, a Bayesian contingency table with the RNG data was computed by collapsing all data into one data set (Table 1). In total, participants (N = 70) named more small numbers (1–4) than large numbers (6–9; distribution: 14032 vs 13248) with the proportion of 0.51 to 0.49 resulting in a $BF_{10} = 593.26$ (very strong evidence) in favour of the alternative hypothesis, indicating a small number bias

Table 1. Absolute frequencies of randomly produced numbers in each direction (excluding the number five).

| Numbers | Direction | | | | | Total |
|---|---|---|---|---|---|---|
| | Baseline | Left | Right | Down | Up | |
| Large (6–9) | 2615 | 2709 | 2638 | 2695 | 2591 | 13248 |
| Small (1–4) | 2782 | 2836 | 2862 | 2782 | 2770 | 14032 |
| Total | 5397 | 5545 | 5500 | 5477 | 5361 | 27280 |

which is a typical RNG signature [75]. However, there was no effect of direction on number magnitude: neither horizontally (left vs. right) $BF_{01} = 27.05$, nor vertically (up vs. down) $BF_{01} = 27.43$, nor when comparing each direction to the baseline: Left vs. baseline $BF_{01} = 38.22$, right vs. baseline $BF_{01} = 36.58$, down vs. baseline $BF_{01} = 30.58$, up vs. baseline $BF_{01} = 41.08$. All these Bayes factors provide strong to very strong evidence in favour of H0, indicating that the isometric directional force failed to induce SNAs in any direction.

**1.2 Force during RNG.** The cluster permutation analysis did not reveal time-windows of interest, therefore five time-windows for each directional condition were selected manually (in ms: 50–150, 150–250, 250–350, 350–450, 450–576). The force data were averaged within the selected time-windows and submitted to Bayesian paired samples t-tests (Student *t* distribution) with factors *small* and *large* within each direction. The tests were conducted in the JASP software 0.16.1 [76], using a Cauchy prior width of .707.

The Bayes factors testing small vs. large number magnitudes revealed moderate ($BF_{01} = 2.99$) to strong ($BF_{01} = 26.60$) evidence for the null hypothesis, depending on the time interval (Table 2), favouring the conclusion that there is no systematic difference between small and large number magnitude regarding force magnitude. Visualized representation of the force data is presented for both laboratories independently in Fig 4A and 4B. The S5 File reports the descriptive statistic of all force data from both laboratories.

## 2. SDA

**2.1 Force during SDA.** Symmetrically to the RNG analysis, we also conducted Bayesian paired samples t-tests (Student *t* distribution) with factors *small* and *large* within each direction using a Cauchy prior width of .707. Again, the cluster permutation analysis did not reveal time-windows of interest, therefore six time-windows for each directional condition were selected manually (in ms: 50–150, 150–250, 250–350, 350–450, 400–500). Each trial (Go and No-Go) that remained after data processing was included in the analysis for operand 1, operator and operand 2. However, during the Go trials answers were not analysed because of the substantial noise in the grip force signal created by speech production during the answer.

Table 2. Bayes factors of Bayesian t-tests of the RNG force data for small vs. large numbers.

| Time-windows in ms | $BF_{01}$ | | | |
|---|---|---|---|---|
| | Left | Right | Up | Down |
| 50–150 | 24.09 | 9.33 | 14.61 | 8.81 |
| 150–250 | 26.60 | 2.99 | 19.56 | 14.98 |
| 250–350 | 23.86 | 6.85 | 22.56 | 10.38 |
| 350–450 | 20.57 | 17.19 | 20.77 | 14.19 |
| 450–576 | 24.21 | 19.67 | 14.19 | 14.82 |

The analysis was performed with n = 69, 66, 63, and 67 for the directions left, right, up and down, respectively.

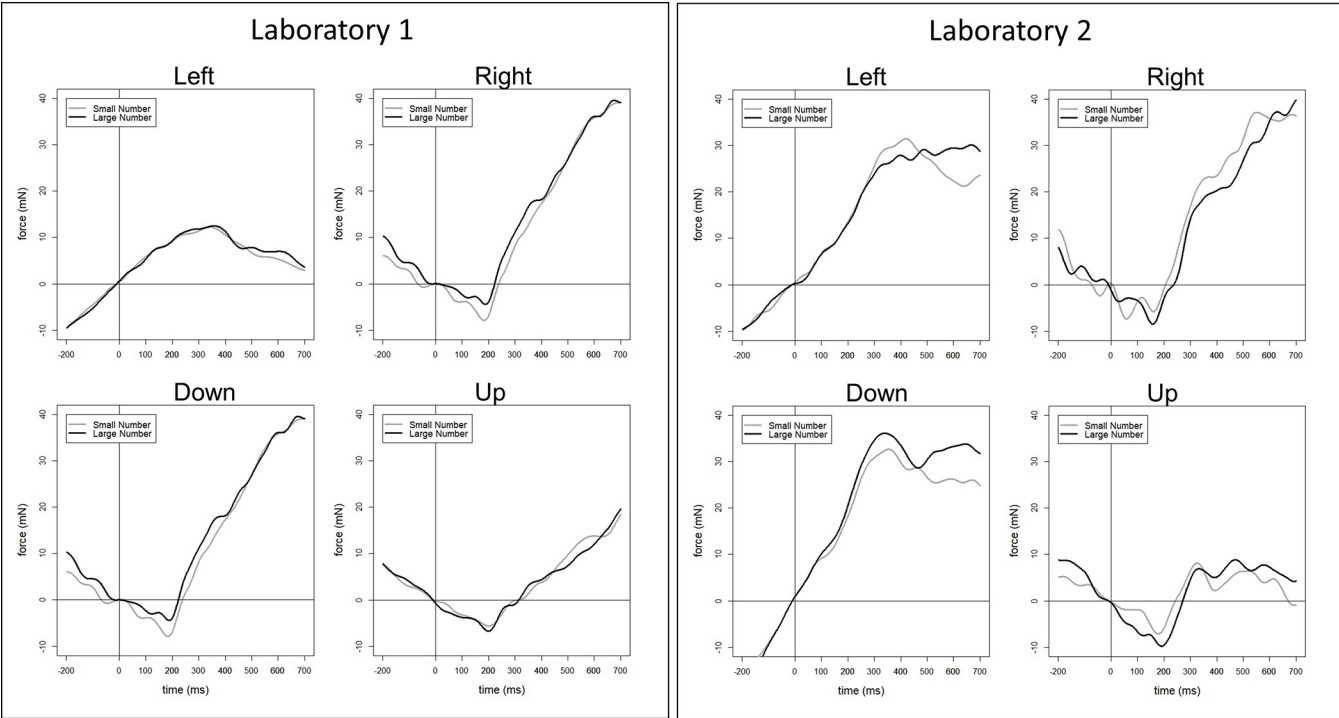

**Fig 4.** A. Visualized Force Data during the RNG Task from Laboratory 1. B. Visualized Force Data during the RNG Task from Laboratory 2. Fig 4A and 4B visualize force profiles of the RNG experiment. The grey lines represent force elicited by a small number, the black lines represent force elicited by a large number, in the respective direction.

Tables 3 and 4 summarize the results. Visualized representation of the force data is presented for both laboratories independently in Fig 5A and 5B. The S5 File reports the descriptive statistic of all force data from both laboratories.

## Discussion

In this study, isometric directional force was implemented as an experimental manipulation in two numerical tasks that required producing continuous isometric force. The first objective was to study if isometric directional force would suffice to produce SNAs across the horizontal and the vertical axes during RNG. This question was motivated by prior studies describing how movements through space influence RNG (e.g., [56,58–60]). Notable, our isometric

**Table 3. Bayes factors of Bayesian t-tests of the SDA force data for a small vs large operand 1.**

| Time-windows in ms | $B_{01}$ Operand 1 | | | |
|---|---|---|---|---|
| | Left | Right | Up | Down |
| 50–150 | 1.82 | 8.02 | 22.78 | 15.02 |
| 150–250 | 7.99 | 9.62 | 24.39 | 3.12 |
| 250–350 | 13.87 | 8.47 | 20.87 | 7.54 |
| 350–450 | 18.32 | 5.73 | 15.18 | 5.81 |
| 400–500 | 19.47 | 8.59 | 10.39 | 7.17 |

The Operand analysis was performed with n = 61, 60, 67, and 65 for the directions left, right, up, and down, respectively.

**Table 4. Bayes factors of Bayesian t-tests of the SDA force data for operator (plus vs. minus), operand 2 (small vs. large), and answer (small vs. large).**

| Time-windows in ms | BF01 | | | | | | | |
|---|---|---|---|---|---|---|---|---|
| | Small O1 followed by: | | | | Large O1 followed by: | | | |
| | Operator | | | | Operator | | | |
| | Left | Right | Up | Down | Left | Right | Up | Down |
| 50–150 | 14.94 | 1.82 | 11.91 | 20.97 | 16.14 | 22.40 | 4.13 | 2.33 |
| 150–250 | 24.20 | 3.43 | 10.94 | 21.10 | 16.02 | 10.40 | 4.40 | 3.72 |
| 250–350 | 18.24 | 1.38 | 11.45 | 21.59 | 16.51 | 8.88 | 1.70 | 7.97 |
| 350–450 | 17.78 | 0.50 | 12.33 | 24.49 | 24.31 | 6.21 | 0.87 | 6.08 |
| 400–500 | 19.95 | 0.77 | 13.40 | 24.21 | 24.59 | 8.78 | 2.26 | 4.74 |
| | O2 | | | | O2 | | | |
| | Left | Right | Up | Down | Left | Right | Up | Down |
| 50–150 | 20.99 | 11.89 | 17.96 | 21.58 | 17.90 | 15.34 | 13.98 | 8.27 |
| 150–250 | 23.52 | 15.57 | 17.33 | 20.41 | 15.32 | 12.88 | 8.72 | 13.41 |
| 250–350 | 24.21 | 8.10 | 18.31 | 23.78 | 17.63 | 16.78 | 7.49 | 18.66 |
| 350–450 | 23.44 | 10.10 | 14.06 | 23.00 | 21.10 | 17.59 | 13.62 | 17.84 |
| 400–500 | 23.60 | 11.00 | 12.74 | 23.92 | 22.04 | 16.32 | 13.68 | 16.74 |
| | | | | | | | | |
| | Answer | | | | Answer | | | |
| | Left | Right | Up | Down | Left | Right | Up | Down |
| 50–150 | 0.69 | 15.21 | 9.72 | 17.41 | 14.34 | 2.65 | 0.02 | 3.96 |
| 150–250 | 0.39 | 15.03 | 7.77 | 22.00 | 13.37 | 1.03 | 0.31 | 4.94 |
| 250–350 | 0.71 | 16.17 | 6.62 | 23.90 | 16.93 | 1.15 | 0.90 | 9.00 |
| 350–450 | 0.16 | 17.86 | 7.65 | 22.74 | 15.38 | 0.62 | 1.02 | 8.42 |
| 400–500 | 0.13 | 17.86 | 8.56 | 22.45 | 20.54 | 1.10 | 0.81 | 7.78 |

The operator sub table compares plus vs. minus after either a small or a large operand one. O2 and the result sub-tables report the results of small vs. large numbers within referred time-windows after either a small or a large operand 1. The Operator analysis was performed with n = 60, 57, 62, and 63 for the directions left, right, up, and down, respectively. The O2 analysis was performed with n = 60, 52, 55, and 62 for the directions left, right, up and down, respectively. The Answer analysis was performed with n = 59, 43, 46, and 60 for the directions left, right, up, and down, respectively.

directional force manipulation did not include a movement through space. Instead, isometric directional force utilized a subtler approach of directionally applied pressure attempting to elicit spatial codes. The second objective was to study if number magnitude would affect force production [54,55].

In the RNG task, our experimental manipulation of isometric directional force failed to elicit SNAs along both axes, indicating that the directional force failed to induce SNAs in any direction. Contrary to the prediction of H1, participants did not generate more small numbers in the left/down condition or more large numbers in the right/up condition. In fact, there is strong evidence that the isometric directional force conditions did not differ from the baseline condition in which participants produced numbers in a relaxing position. Nevertheless, participants displayed a typical small number bias [75] indicating that the RNG task, by itself, was carried out appropriately.

Overall, this null result is likely due to the design choices of the experiment. Particularly, the experiment was designed to exclude task-specific spatial influences and instead utilized directional (i.e., isometric) force. The central hand position probably eliminated those spatial references, and therefore prevented SNAs [30,31]. The isometric directional force by itself was then insufficient to activate spatial references and, thus, produce SNAs. Excluding spatial task

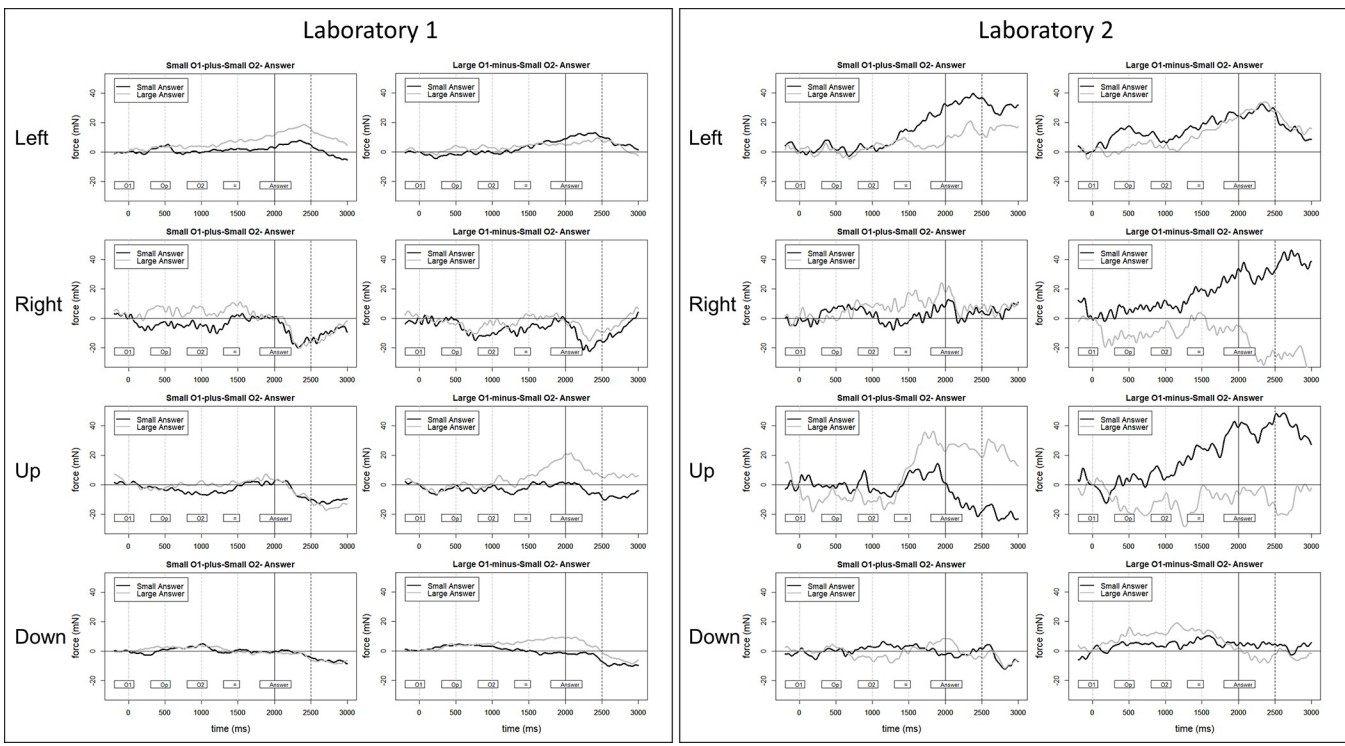

**Fig 5.** A. Visualized Force Data during the SDA Task from Laboratory 1. B. Visualized Force Data during the SDA Task from Laboratory 2. Fig 5A and 5B visualize force profiles in the SDA experiment. Representation of a full trial that compares small to large answers. Hereby, there exist only two possible combinations that lead to small and large answers. The grey and black lines represent force elicited in specific conditions (Grey line: Small operand 1 plus small operand 2, and small answer: Black line: Large operand 1 minus small operand 2, and large answer). The vertical dotted line represents the onset of a specific part of the equation (operand 1, operator, operand 2, equals, or the answer, respectively).

demands is atypical for a study accessing SNAs (see Figs 1 and 2) and follows the logic of the falsification by elimination approach [61]. In this experiment, we therefore systematically excluded known spatial factors under which prototypical SNAs studies are conducted.

Our finding is in contrast to other RNG studies that found SNAs along the horizontal and vertical axes by using either active movement ([59]; see also [56] for RNG across the horizontal axis; for review see [28]) or passive movement through space [60]. Therefore, it is likely that activating a spatial code in the response space, it requires a movement with a spatial component, such as movement through space or movement allocated in space (e.g., contacting spatially aligned response buttons), to create SNAs. Interestingly, theoretical accounts described in the Introduction, such as the reading and writing directions [2], hemispheric asymmetry, serial working memory [8], the polarity correspondence [9,10], and the embodied cognition accounts (e.g., [13,29]) all make no predictions for cases where movement in space is not present. Importantly, movement in space is often measured as a response (e.g., manual response, gaze shift) or executed before the response (RNG paradigm, e.g., [56]). Therefore, spatial information (e.g., movement through space) seems to be deeply rooted into SNAs assessment. To investigate the role of movement in space, future experiments could contrast designs utilizing movement in space against designs that utilize static and central versus static and lateral responses (for static and lateral responses see [41]).

While isometric directional force failed to induce SNAs, H2 predicted a reverse influence such that number magnitude affects force magnitude [54,55,58]. However, the average Bayes factors provide moderate to strong evidence that number magnitude had no effect on force

magnitude. Specifically, in the RNG task, the intention of producing either small or large numbers had no effect on force magnitude. Apparently, the mechanisms involved in random production of numbers either did not induce changes in isometric force or we could not capture them under the current experimental design. This being said, we should bear in mind that we only analysed the force data before voice onset; this was necessary to prevent artefact contamination by the speech act. However, given that we relied on the mean voice onset, it cannot be ruled out that the last 100–200 ms were already in many trials part of the speech production phase [77] and did not only reflect numerical processing.

Another research question of the experiment was to explore SNAs on a temporal continuum during mental arithmetic. Hereby, number magnitude also did not affect force magnitude in the SDA task systematically at any time point during sequential presentation of both operands, the operator, and during the answer. Notably, there are two exceptions during the processing of the answer. First, there is moderate evidence of an effect (BF01 = 0.16 and 0.13 translating to BF10 = 6.25 and 7.69) while processing the answer (350–450 and 400–500 ms after the onset of the answer) following a small operand 1 in the direction left. However, inspecting the visual data of the individual laboratories (see Fig 5A and 5B) shows an opposite effect of number magnitude on force magnitude. Laboratory 1 shows larger forces for larger answers while Laboratory 2 shows larger forces for smaller answer. Second, similar evidence is found while processing an answer following a large operand 1 (50–150 and 150–250 ms) in the direction up. Here, we report strong to moderate BF01 = 0.02 and 0.31 (translating to BF10 = 50 and 3.33), respectively. Again, inspecting the visual data of the individual laboratories (see Fig 5A and 5B) shows an opposite effect of number magnitude on force magnitude with Laboratory 1 showing larger forces for larger answers and Laboratory 2 showing larger forces for smaller answer. Both of these asymmetrical and inconsistent effects could have been the result of different sample sizes. The data collection of Laboratory 2 was stopped before fulfilling the a priori calculated sample size requirement of N = 52. Apart from these exceptions, there was no main effect of numbers and operator: neither large numbers/nor the operational sign affected force production.

Overall, Bayes factors during the SDA task were not as consistent as during the RNG task (SDA task: BF01 range: 0.02–24.59 and RNG task: BF01 range: 2.99–26.6). One plausible explanation would be that the mental arithmetic procedure in itself activated some spatial information [78], however, not strong enough to elicit reliable magnitude activation in isometric force. This could be attributed to the habitual reading of arithmetical tasks on paper which are read from left to right.

Theoretically, the justification of a null effect is challenging because no such conclusion is logically permitted. There always will be the possibility that there is an effect which we failed to find. Therefore, and to better describe the null results, the Bayesian framework was chosen over the Frequentist approach. Specifically, there are two main reasons to apply the Bayesian framework. First, it allows for the quantification of the amount of evidence in favour or either the null or the alternative hypothesis. This approach does not force us to decide whether there is an effect or not as the Frequentist framework does. Instead, the Bayes factors indicate how much evidence there is in favour of a hypothesis.

Second, the practical reason is how Bayes factors allow to combine data sets. For this experiment, we have collected data across two laboratories with similar set-ups, identical code and study protocols. Our objective was to provide converging evidence by replicating findings across laboratories. Yet, combining the datasets might be a suboptimal procedure due to small technical differences between laboratories. For example, the soft- and hardware of the computers could produce timing differences that systematically apply to one specific laboratory. Therefore, we chose to produce two individual result tables (see S1 File) and then sum them

up to one by multiplying the Bayes factors. This logic is based on Bayesian knowledge updating and how data sets are used to inform each other using prior probability [73].

## Limitations

In this study, directional force generation was ensured by measuring isometric force production. For the experimental manipulation to induce SNAs by directional force, it was therefore sufficient to press the force sensor into an instructed direction (see Fig 4A and 4B). Indeed, as previously shown, an isometric force manipulation (but not measurement) during a lateral movement was successfully applied to induce spatial biases across the horizontal axis in a mathematical problem-solving task (Experiment 2 in [79]). Participants generated more addition solutions following a right-sided movement and more subtraction solutions following a left-sided movement. Notably, these authors relied on lateral movements along the surface of a touch-screen, meaning that participants pressed the touch-screen while conducting a lateral movement.

However, there can be the case that isometric force as a measurement lacks sensitivity to reflect cognitive processes. There are two counterarguments to this statement. First, visually observing our own force data of the SDA task (see Fig 5A and 5B) indicates a sudden drop after the presentation of the answer (2000 ms). We suggest that this drop in isometric force represents a decrease in cognitive load [80] after the arithmetic procedure is completed. Second, as stated in the introduction, Miklashevsky et al. [41] found SNAs in numerical tasks during bimanual and passive isometric force recordings. The authors employed an experimental design utilizing two lateralized force sensors in a task that did not require active responses and instead recorded passive forces applied to the sensors. The authors found that smaller numbers induced more force in the left hand and larger numbers induced more force in the right hand. Such a finding is particularly intriguing because SNAs were produced without an active response and therefore not relying on a movement through space. This finding suggests that movements through space are not necessary in order to elicit SNAs. Instead, spatial information about effector positions by itself might be the sole contributor to SNAs formation (cf. [81]). Indeed, the authors utilized two spatially aligned sensors and did not control for spatial information in the response space.

Additionally, keep in mind that Miklashevsky et al. [41] employed two different tasks compared to our study. The authors themselves suggested that the task itself is critical in finding SNAs as they report weaker SNAs in their surface numerical decision task ("decide if this a number or a letter") than in the classical magnitude classification task ("is this number larger or smaller than 5"). Therefore, it is an open empirical question how their finding will translate to other experimental tasks such as random number generation and mental arithmetic as was the case in the current study. Together, Miklashevsky et al.'s [41] study and our own can be interpreted as suggesting that active responses (and movements) might not be per se necessary to elicit SNAs, while spatial information might play a critical role. Future experiments could test more systematically if SNAs can exist in an experimental design space that excludes both, spatial information in the presentation and response space, as well as movements in the response space. Therefore, the current study contributes to the overarching discussion on design space and effect specification [82] necessary to elicit SNAs.

Another limitation is a substantial difference between Miklashevsky et al.'s [41] and the current study is <u>how</u> the other authors employed the force sensor, namely in a manual precision grip. Adjusting one's precision grip so the object does not slip, reflects exquisite fine motor control [83]. However, the precision grip is motorically different from the rather coarse directional isometric conditions employed in the current study. In contrast to the precision grip, the current study utilized the crude motor capabilities of participant's fingers, especially the

index and the middle fingers. It is likely that these fingers do not produce such finely tuned force patterns as is typical for the precision grip. Indeed, recently, it has been shown that the magnitude of unintentional force drift is large for individual fingers, especially the index finger [84]. Additionally, visually, we can observe the pattern of force in different directional conditions and how it changes drastically between directions (see Figs 4A, 4B, 5A and 5B). We attribute this to the asymmetrical motor difficulty between force directional conditions of left/ down and right/up. Especially the right/up conditions, in contrast to the left/down conditions, are suffering from a larger attrition rate due to motor difficulty. This is unfortunate as the right/up directions should be associated with larger numbers. However, both left/ down directions suffer less from artefact rejection and yet present average Bayes factors pointing at moderate evidence in favour of a null effect regarding numerical magnitude processing. At the very least, isometric force served as a manipulation in any direction and ensured that participants deployed directional force (see limitations of Experiment 2 in [79]). Yet, in our independent assessment of H1, we do not find any SNAs in any direction during the RNG task.

Alternatively, the precision grip can be associated with purpose and affordances [85] as it is real-life relevant and frequently used for fine-motor activities such as holding a pen. This is also true for the field of numerical cognition where multiple studies found that small numerical values facilitate a precision grip, while large numerical values facilitate a power grip [86–88]. In the current study, the experimental task by itself might have not been relevant to activate affordances for the directional force that resulted in the absence of SNAs. This is also in line with the embodiment approach according to which movements become only meaningful when they are goal-directed (e.g., [89]).

Another limitation can be found in the SDA task itself. The process of mental arithmetic presupposed showing several subsequent numbers connected by an operator. This chain of stimuli might have resulted in accumulation of response biases to separate stimuli, making it difficult to disentangle the effects of each component. Further studies could display stimuli with a longer duration to increase the visibility of overlapping effects as a possible solution to the problem.

## Conclusion

Studies that find spatial-numerical associations usually include spatial information in their experimental design. Such spatial information can be found in lateralized presentation of stimuli, visual directional cueing, laterally arranged response buttons and movement in space. The current study has eliminated such spatial information by centrally aligning the space of presentation and response. The aim was to produce SNAs with isometric force applied in the directions along the horizontal and vertical axes, thus removing movement and lateralized spatial information. Such isometric directional force was insufficient to produce SNAs along the horizontal and the vertical axes during the RNG task. Therefore, our motor task did not interact with the concept of magnitude. Additionally, we explored whether number magnitude affects force magnitude in such a design. Again, we report null results for such magnitude and force interactions. Therefore, participants' concept of magnitude did not interact with the motor task. We conclude that applying pressure into the direction of left, right, up, and down does not suffice in eliciting SNAs along the horizontal and the vertical axes.

## Supporting information

**S1 Table. Experimental factors of the SDA task.** S1 Table shows the experimental factors of the SDA task.
(DOCX)

**S2 Table. Experimental stimuli of the SDA experiment.** S2 Table shows the experimental stimuli of the SDA task. In total, 58 math problems (48 true and 10 distractors) were used as stimuli. The stimuli were presented twice across four blocks. Number magnitude was defined as small (numbers 1,2,3,4) and large (6,7,8,9). Number 5 was excluded from both operands and the result.
(DOCX)

**S1 File. Data from individual laboratories.**
(DOCX)

**S2 File. Calibration procedure.**
(DOCX)

**S3 File. Additional information for the analysis.**
(DOCX)

**S4 File. Pilot studies.**
(DOCX)

**S5 File. Descriptive statistics of force data from both laboratories.**
(DOCX)

## Acknowledgments

We would like to thank Andre' Dähne for his graphical support. We also would like to thank Jan Kaminski, Dominik von Hertlein, Tobias Nolte, and David Voß for their contributions in data collection.

## Author Contributions

**Conceptualization:** A. Michirev, K. Kühne, M. H. Fischer, M. Raab.

**Data curation:** A. Michirev, K. Kühne.

**Formal analysis:** A. Michirev, K. Kühne.

**Funding acquisition:** M. H. Fischer, M. Raab.

**Methodology:** A. Michirev, K. Kühne, O. Lindemann, M. H. Fischer, M. Raab.

**Project administration:** M. H. Fischer, M. Raab.

**Resources:** M. H. Fischer, M. Raab.

**Supervision:** M. H. Fischer, M. Raab.

**Validation:** A. Michirev, K. Kühne, M. H. Fischer, M. Raab.

**Visualization:** A. Michirev, K. Kühne.

**Writing – original draft:** A. Michirev, K. Kühne.

**Writing – review & editing:** A. Michirev, K. Kühne, O. Lindemann, M. H. Fischer, M. Raab.

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
