## [Decision Letter · Decision Letter 0]

7 Feb 2023

PONE-D-22-30799How to not induce SNAs: the insufficiency of covert directional movementPLOS ONE

Dear Dr. Kühne,

Thank you for submitting your manuscript to PLOS ONE. After careful consideration, we feel that it has merit but does not fully meet PLOS ONE’s publication criteria as it currently stands. Therefore, we invite you to submit a revised version of the manuscript that addresses the points raised during the review process.

We look forward to receiving your revised manuscript.

Kind regards,

Alessia Tessari, Ph.D.

Academic Editor

PLOS ONE

Journal Requirements:

2. Please provide additional details regarding ethical approval in the body of your manuscript. In the Methods section, please ensure that you have specified the name of the IRB/ethics committee that approved your study.

Additional Editor Comments:

Dear Dr Kühne,

We have now received two reviews. After reading the paper myself, I found it interesting for PLOS ONE, but before considering it for publication, you have to arrange the manuscript following the reviewers' requests.

I look forward to receiving your revised version of the manuscript.

Sincerely,

Alessia Tessari

Reviewers' comments:

Reviewer's Responses to Questions

**Comments to the Author**

1. Is the manuscript technically sound, and do the data support the conclusions?

Reviewer #1: Yes

Reviewer #2: Partly

2. Has the statistical analysis been performed appropriately and rigorously? 

Reviewer #1: Yes

Reviewer #2: Yes

3. Have the authors made all data underlying the findings in their manuscript fully available?

Reviewer #1: No

Reviewer #2: No

4. Is the manuscript presented in an intelligible fashion and written in standard English?

Reviewer #1: Yes

Reviewer #2: Yes

5. Review Comments to the Author

Reviewer #1: The present paper reports a study in which participants are asked to maintain a constant force with a single hand on a disk plate in one of four directions: upwards, downwards, leftwards or rightwards, while they carry out a random number generation task and an arithmetic verification task. No movement was produced, only isometric force. In the number generation task they measured the magnitud of the generated numbers as a function of instructed force direction, and they also measured the magnitud of the force generated as a function of the magnitud of the number generated. In the arithmetic verification task they measured the magnitud of force produced as a function of the magnitud of each operand, the type of operation (addition vs. subtraction), and the magnitud of the result. The goal was to detect Space-Number Associations (SNAs), such that left or up space is associated to small numbers and right or low space is associated to large numbers. In all cases, they failed to find any significant SNA. Bayesian analyses showed that there was clear support for the null hypothesis. The authors interpret that it is the lack of outer directional movement what is causing the absence of SNAs.

The paper is very well written, the topic is relevant, the methods are sound, the data are clear and clearly support the null hypothesis and the interpretation follows from the context and the data. All in all, I have very few concerns or suggestions for improvement, and I do recommend its publication after improving on those points.

My questions and suggestions are the following:

- PLoS ONE requests that the data be openly available and the authors claim that they will be uploaded to an open repository after the acceptance of the manuscript. However, in the Data Availability Statement at the end of the manuscript the authors say that data are “available upon reasonable request from the corresponding author”. In my opinion, this goes against the open data policy of the journal. In my experience, asking authors for data of published studies succeeds in very rare occasions. The data should be deposited in a open repository such as OSF, where they can be directly accessed in the future without any intervention from the authors.

- I don’t understand why the hypotheses on the random number generation task are posed as confirmatory and those on the arithmetic verification task are posed as exploratory. Lacking a preregistration of the study (as it is the case), all of them should be considered exploratory.

- I also don’t understand why, if the authors have examined the interaction between number magnitud and the four force directions, they assert that their design includes the factor force direction with two levels “(left/right, up/down)”. If they carried out independent analyses for the two levels within each axis (horizontal and vertical), they should say that there were two independent designs. If they included all four directions into a single design, the factor has four levels.

- The preprocessing and analysis of force data is complex and implies a high number of choices. As a suggestion for future research, I would be more convinced if every choice would be pre-registered.

- I would like the authors to justify their choice of prior in the Bayesian analyses (line 506).

- My MAIN SUGGESTION for improvement is to include in the Discussion the recent study by Miklashevsky et al. (2022), who found SNAs using grip force and without any overt movements. This study seems to clash with their conclusion that it is the lack of overt movements what is responsible from the absence of SNAs, and should be taken into consideration and provide some speculation as to why the contrast between the studies.

- Is citation number 73 correct? (line 655). The cited study does not measure any forces.

- In the paragraph starting in line 667, the authors entertain a possible explanation of the absence of effects in the present study that strikes me as undermining the whole study: they say that perhaps the kind of force that they measured (pushing with two fingers in a given direction) may not be sensitive to high level cognitive processes in the way that force in a precision grip has been already shown to be. If this is so, the authors are saying that they are using a measure that is invalid as an index of the cognitive processes of interest. They should strive to rule out this possibility, either with good arguments or by carrying out an experimental construct validation of the measure. If the authors finally accept that they cannot be sure that the measure is a valid one, then they should change their conclusions: the SNAs are absent either because there is no overt directional movement or because the measure is not valid.

References:

Miklashevsky, A., Fischer, M. H., & Lindemann, O. (2022). Spatial-numerical associations without a motor response? Grip force says ‘Yes’. Acta Psychologica, 231, 103791. https://doi.org/10.1016/j.actpsy.2022.103791

Reviewer #2: There are several unique merits to the paper, not the least of which is its creative methodological approach. The paper is mostly clear (with one exception, see below), and I see no problems with the design or analysis. My main concern(s) stem(s) from the conclusions that are reached based on the results. Below, I elaborate on what I see as the central concern(s).

1. It seems that the central argument relies on the interpretation of a null effect. Of course, there are some instances where null effects are valuable, and the statistics were appropriate given the sort of conclusion the authors wished to make. Nevertheless, I had trouble making sense of this particular null effect, because I have no standard upon which to evaluate it. Are there cases where we *should* expect this method to reveal SNAs? If I'm understanding correctly, this approach has only been used successfully in the study of language. I'm not an expert on those studies or this methodology, but it seems that even the authors agree their approach is somewhat different from what is used in prior work. Therefore, I would be uncomfortable concluding that covert directional movements *cannot* induce SNAs if we do not have strong reason to believe that this design is sufficiently sensitive to detect covert directional movements to the relevant extent. In other words: I feel that we'd need a comparable positive effect in order to evaluate the lack of an effect here. Is there anyway to demonstrate that this method *is* sensitive enough to detect some effect that we'd have strong reason to predict?

2. Is there something circular about the central argument? The authors argue that explicit movement is required to induce SNAs. But isn't it also possible that this measure is just not a measure of SNAs? In other words: Am I correct to understand that there is no independent evidence of SNAs in this task? If that's true, is it not possible that participants simply exhibited no SNAs, but that, if they had, this measure may have correctly detected them?

In other words, there are two possible conclusions one could draw from the null effect. One conclusion is that subjects are not exhibiting SNAs. But another equally valid conclusion is that the subjects are exhibiting SNAs, but that this measure isn't sufficient to detect it. Perhaps, for instance, eye movements could have revealed SNAs even if these motor movements did not.

3. In the arithmetic task, how are the equations appearing? The language in the Methods is not clear. It says the stimuli were presented "sequentially", but where? Were these stimuli arranged spatially in any way, or did they all appear in the same location one after the other? If the latter, what was the timing of that like?

Ultimately, I think it could be valuable to have this result published. Perhaps others would be inspired to take a similar approach, which could prove valuable. However, I feel that the results should not be published without, at the very least, clearly qualifying the results. I'm just not sure that we could take from a single sort of null effect, in a very untested paradigm, that anything certainly is or certainly is not true. Then again, the authors' may have some very strong argument as to why they think these results do merit the conclusions that they made. Or it may be that I am misunderstanding something about the study. If either of these things were true, I would be able to more enthusiastically recommend publication.

Other:

1. I found it surprisingly difficult to understand the key manipulation. I spent much time wondering whether the experiment involved applying pressure to the subjects' hands, or whether the authors were measuring covert movements -- or, both. I'm still not sure. I think this stems from the ambiguity in phrases like "We studied continuous isometric forces..." It just wasn't clear to me whether the 'forces' referred to a force that was being applied by the device, or by the participant. I think this problem can be easily solved by just revisiting this language through the manuscript, to ensure that it would be clear to a reader that has no prior knowledge of the procedure or design.

2. I had trouble understanding what counts as 'overt' directional movement vs. 'covert' directional movement. In the classic SNARC design, is there overt directional movement? Subjects are just keeping their hands in one place. Why would this be overt? And here, if forces are being applied in one direction or the other, isn't that quite 'overt'? It's an explicit manipulation that the subjects would ostensibly be aware of. There are several other border cases that I'm just not sure about. For instance, there are a few papers from Stella Lourenco's lab that seem to fall somewhere in between:

Holmes, K. J., Ayzenberg, V., & Lourenco, S. F. (2016). Gamble on gaze: Eye movements reflect the numerical value of blackjack hands. Psychonomic Bulletin & Review, 23, 1974-1981.

Aulet, L. S., Yousif, S. R., & Lourenco, S. F. (2021). Spatial–numerical associations from a novel paradigm support the mental number line account. Quarterly Journal of Experimental Psychology, 74(10), 1829-1840.

Aulet, L. S., & Lourenco, S. F. (2018). The developing mental number line: Does its directionality relate to 5-to 7-year-old children’s mathematical abilities?. Frontiers in Psychology, 9, 1142.

In the first, eye movements alone were used as a measure of SNAs. In the second and third, placement errors in a spatial task were used as a measure of SNAs. The latter certainly required movement, but of a very different kind that in many SNARC paradigms. The former obviously involves a kind of movement, but surely the present study also involves eye movements -- so shouldn't SNAs be induced by that, regardless of the other manipulation?

I don't have any strong opinions about this, except that, having read the paper several times, I'm not sure what counts as covert vs. overt.

3. Related to both points above: If the manipulation involves instructing participants to apply forces in different directions, how is that not overt? Sorry to be difficult about all of this; I find myself very confused.

Minor:

1. The first sentence of the abstract doesn't stand on its own. What does it mean that people "respond faster to the left", for instance? Respond to what?

6. PLOS authors have the option to publish the peer review history of their article (what does this mean?). If published, this will include your full peer review and any attached files.

Reviewer #1: **Yes: **Julio Santiago

Reviewer #2: No

---

## [Author Response · Author response to Decision Letter 0]

22 Mar 2023

Reviewer #1:

1. PLoS ONE requests that the data be openly available and the authors claim that they will be uploaded to an open repository after the acceptance of the manuscript. However, in the Data Availability Statement at the end of the manuscript, the authors say that data are “available upon reasonable request from the corresponding author”. In my opinion, this goes against the open data policy of the journal. In my experience, asking authors for data from published studies succeeds on very rare occasions. 

Reply:

Thank you for this reasonable suggestion. We have now deposited our data and all the analyses scripts in an open repository (OSF, https://osf.io/v7dyj/) and corrected our Data Availability Statement correspondingly (lines 887-888).

2. I don’t understand why the hypotheses on the random number generation task are posed as confirmatory and those on the arithmetic verification task are posed as exploratory. Lacking preregistration of the study (as is the case), all of them should be considered exploratory.

Reply:

Thank you for this comment. Indeed, we did not preregister any of the hypotheses of this experiment. We explicitly state this now in lines 325-327: “The following hypotheses were not preregistered but formulated a priori and derived from theory and previous empirical work.”. Additionally, we removed Hypothesis 3 and reframed it as a research question exploring potential effects. In lines 340-343 it now reads “As an additional research question of the experiment, we aimed to explore SNAs on a temporal continuum during mental arithmetic, thus testing how activation of number concepts affects force production during mental arithmetic.”. We have also explicitly stated that the analysis method also was not preregistered but generally followed the guidelines proposed by Nazir et al. (2017). In lines 441-442 it now reads “The following method was not preregistered but closely followed the recommendations of Nazir et al. [61, for details see supplementary file S5]. ”.

As a general comment, we want to clarify that our RNG hypotheses are both strongly derived from and predicted by empirical and theoretical work. Therefore, we had clear hypotheses we put to test. To make it clearer, we have highlighted the bi-directionality of concept-motor interaction before introducing the first two hypotheses. In line 324-325 that now reads “This approach allows to test the bi-directionality of the concept-motor interactions.”. Additionally, half sentences were added to H1 and H2 that specify the concept-motor directions and now reads as (lines 328-332) “H1. We hypothesize that covert directional movement activates spatial codes that affect the production of random numbers, thus testing how (covert) movement affects the conceptual representation of numbers. That is, participants are expected to produce more small numbers when pressing in the direction left/down and more large numbers when pressing in the direction right/up [51,54].” And (lines 333-339) “H2. We hypothesize that generating numbers by itself affects motor activation [53], thus testing how conceptual representation of numbers affects force production during covert movement. Generating numbers is assumed to activate within-magnitude associations [ATOM, 49,50], which in turn, are hypothesized to modulate the amplitude of applied isometric force. Hereby, generating smaller numbers should produce smaller force while generating larger numbers should produce larger force when simultaneously pressing against the surface of a force sensor [49,50].”. 

Similarly, the research question (original H3) is also based on the theoretical predictions of A Theory of Magnitude (Walsh, 2003; 2015). However, the important difference is that the timing of the effect is theoretically unclear and the empirical evidence is rather mixed. Therefore, we did not specify it. We have improved the section that now reads (lines 340-353):

“As an additional research question of the experiment, we aimed to explore SNAs on a temporal continuum during mental arithmetic, thus testing how activation of number concepts affects force production during mental arithmetic. The rationale is similar to that of the target detection paradigm (see Fig 1, example 6). However, we removed spatial information from the presentation and the response space. Therefore, instead of providing spatial information in the form of targets to probe these SNAs, we again relied on central stimuli presentation and covert directional force. In this task, spatial associations of numbers could become activated in relation to the arithmetic procedure; namely either during the first operand, or the operation sign, or the second operand, or during the answer. Similarly to the reasoning in H2, number meaning related to the task should be expressed by the participants’ spontaneous force production, with smaller numbers generating smaller force and large numbers generating larger force [49,50].”. 

Further, we fully agree that pre-registering is a better scientific practice. We have not preregistered this experiment because it is the first in a larger project that helps us to accumulate knowledge about our experimental design and method. Further experiments within this project will be preregistered. 

3. I also don’t understand why, if the authors have examined the interaction between the number magnitude and the four force directions, they assert that their design includes the factor force direction with two levels “(left/right, up/down)”. If they carried out independent analyses for the two levels within each axis (horizontal and vertical), they should say that there were two independent designs. If they included all four directions into a single design, the factor has four levels.

Reply:

Thank you for this valuable notice. We have changed it accordingly and specified that our design presupposed two independent axes (horizontal and vertical). Changes to this effect appear in the General Method section in lines 388-389: “A customized home-made wooden/metal box was built to enable covert directional movements (i.e., participants’ exerting isometric forces on a fixed force sensor surface) along the vertical and the horizontal axes (two levels each: left/right on a horizontal axis or up/down on the vertical axis).” Also edited in lines 416-418:” We have utilized a within-subject design with one within-subject factor being the force direction (two levels each: left/right on a horizontal axis or up/down on the vertical axis).”.

We also made in clearer in the Introduction why we expect the horizontal and the vertical axes to be independent of each other. In lines 100-104 it now reads: ”Vertical associations instead reflect other sensory and motor experiences, such as the growth of piles during object accumulation or the rise of water in a container, which inspired the linguistic metaphors mentioned above. Together, the embodied cognition approach predicts SNAs on the horizontal and the vertical axes based on different and independent mechanisms. ”. We also extended this argument in lines 240-243 that now reads “One possible explanation for why vertical SNAs remained while horizontal SNAs disappeared could be the hierarchical nature of body-related knowledge representations involved in the task, according to which different mechanisms induce different types of SNAs, referred to as grounded, embodied, or situated [28]. ”.

4. The preprocessing and analysis of force data is complex and implies a high number of choices. As a suggestion for future research, I would be more convinced if every choice would be pre-registered.

Reply:

Thank you for the comment. We agree that in future research we will pre-register the method. To clarify, we have rigorously followed the general guidelines of an existing methodological paper describing the same method we have used (Nazir et al., 2017). We have now made it clearer by adding the reference in the “pre-processing force data” section (lines 542-543). Now it says “Pre-processing of force data followed the general guidelines proposed by Nazir et al. [61] and is documented in the S5 file.” Additionally, we made changes to the Supplemental File (S5, lines 13-16) to clarify that there was one parameter (amplitude range) that we adjusted. We adjusted amplitude range from ±200 to ±400 mN because we utilized longer trials than Nazir et al. [1] that naturally have higher force fluctuation. Specifically, Nazir et al. [1] utilized 1 second trials while we had 2 seconds (RNG) and 2.5 seconds (SDA).” 

References: 

Nazir, T. A., Hrycyk, L., Moreau, Q., Frak, V., Cheylus, A., Ott, L., Lindemann, O., Fischer, M. H., Paulignan, Y., & Delevoye-Turrell, Y. (2017). A simple technique to study embodied language processes: the grip force sensor. Behavior Research Methods, 49(1), 61–73. https://doi.org/10.3758/s13428-015-0696-7

5. I would like the authors to justify their choice of prior in the Bayesian analyses (line 506).

Reply:

Thank you for your suggestion. We have now explicitly justified the choice of the Cauchy prior in the Bayesian analyses (lines 562-565). It now reads: “The default width parameter of .707 was used in all analyses since we did not have strong assumptions about the distribution. Moreover, it was shown that the prior width has little impact on the Bayes factor [70]. All analyses were done within each independent axis (horizontal or vertical).”.

References:

van Ravenzwaaij, D., & Wagenmakers, E-J. (2022). Advantages masquerading as "issues" in Bayesian hypothesis testing: A commentary on Tendeiro and Kiers (2019). Psychological Methods, 27(3), 451-465. https://doi.org/10.1037/met0000415

6. My MAIN SUGGESTION for improvement is to include in the Discussion the recent study by Miklashevsky et al. (2022), who found SNAs using grip force and without any overt movements. This study seems to clash with their conclusion that it is the lack of overt movements that are responsible for the absence of SNAs, and should be taken into consideration and provide some speculation as to why the contrast between the studies.

Reply:

Thank you for the valuable suggestion. Indeed, that paper is highly relevant. We have fully integrated it in our Introduction and Discussion and it enormously helped extending our argument and the limits of the current study and also the validity of our force measure. It also helped us to change and adapt our conclusion on several occasions throughout the manuscript. The main changes in the introduction are in lines 180-189 which now reads: 

“Very recently, a new study challenged the importance of active responses in the response space. Miklashevsky et al. [40] relied on a paradigm of passive and continuous isometric force readings during numerical tasks. Participants produced more force in the left hand while processing smaller numbers and more force in the right hand while processing larger numbers. These results suggest that active responses might not be required to induce SNAs. However, the authors did not control for spatial effects in their design. Specifically, in the response space, the authors utilized two sensors that were held in the left and in the right hand, thus being laterally displaced. Therefore, spatial information in the response space was apparently sufficient to elicit SNAs in their study.“.

In the Discussion, in lines 785-811 it now reads:

“Second, as stated in the Introduction, Miklashevsky et al. [40] found SNAs in numerical tasks during passive isometric force recordings utilizing the same method as in our study. The authors employed an experimental design utilizing two lateralized force sensors in a task that did not require active responses and instead recorded passive forces applied to the sensors. The authors found that smaller numbers induced more force in the left hand and larger numbers induced more force in the right hand. Such a finding is particularly intriguing because SNAs were produced without an active response and therefore not relying on an overt movement through space. This finding suggests that overt movements through space are not necessary in order to elicit SNAs. Instead, spatial information about effector positions by itself might be the sole contributor to SNAs formation [cf. 78]. Indeed, the authors utilized two spatially aligned sensors and did not control for spatial information in the response space.

Additionally, keep in mind that the authors employed two different tasks compared to our study. The authors themselves suggested that the task itself is critical in finding SNAs as they report weaker SNAs in their surface numerical decision task (“decide if this a number or a letter”) than in the classical magnitude classification task (“is this number larger or smaller than 5”). Therefore, it remains to be seen how their finding will translate to other experimental tasks such as random number generation and mental arithmetic as was the case in the current study. Together, Miklashevky et al.’s [40] study and our own can be interpreted as suggesting that active responses (and overt movements) might not be per se necessary to elicit SNAs, while spatial information might play a critical role. Future experiments could test more systematically if SNAs can exist in an experimental design space that excludes both, spatial information in the presentation and response space, as well as overt movements in the response space. Therefore, the current study contributes to the overarching discussion on design space and effector specification [79] necessary to elicit SNAs.”. 

Other adaptations of the conclusion:

1. Lines 31-33 “Here we argue that overt movements in space contribute to the formation of spatial-numerical associations (SNAs).”.

2. Lines 36-37 we have deleted the final sentence. It no longer reads: “We conclude that overt movements through space enable SNAs.”. 

3. Lines 291-293:” Generally, spatial information in the design (e.g., movements through space or responses in space) seems to be essential when investigating SNAs, both as a response producing SNAs and to activate SNAs.”.

4. Lines 300-302:” Considering all the presented evidence, we can conclude that spatial information is omnipresent in many experimental paradigms that study SNAs.”.

5. Lines 697-698: “Therefore, spatial information (e.g., movement through space) seems to be deeply rooted into SNAs assessment. To investigate the role of movement in space, future experiments could contrast designs utilizing movement in space against designs that utilize static and central versus static and lateral responses [for static and lateral responses see 40].”.

6. Lines 866-867: “We conclude that lateralized spatial information in the design space plays a crucial role in eliciting SNAs.”.

References:

Almaatouq, A., Griffiths, T. L., Suchow, J. W., Whiting, M. E., Evans, J., & Watts, D. J. (2022). Beyond Playing 20 Questions with Nature: Integrative Experiment Design in the Social and Behavioral Sciences. Behavioral and Brain Sciences, 1-55. https://doi.org/10.1017/S0140525X22002874

Miklashevsky, A., Fischer, M. H., & Lindemann, O. (2022). Spatial-numerical associations without a motor response? Grip force says ‘Yes’. Acta Psychologica, 231, 103791. https://doi.org/10.1016/j.actpsy.2022.103791

Rizzolatti, G., Riggio, L., & Sheliga, B. (1994). Space and Selective Attention. In Attention and Performance XV. https://doi.org/10.7551/mitpress/1478.003.0016

7. Is citation number 73 correct? (line 655). The cited study does not measure any forces.

Reply:

Thank you for the comment. Indeed, the authors do not report any force measurements. We have now clarified that the study (now not citation number 73 but 76) only manipulated the movement direction and isometric force but did not measure isometric force. In lines 771-779 it now reads: “Indeed, as previously shown, an isometric force manipulation (but not measurement) during a lateral movement was successfully applied to induce spatial biases across the horizontal axis in a mathematical problem-solving task [Experiment 2 in 76]. Participants generated more addition solutions following a right-sided movement and more subtraction solutions following a left-sided movement. Notably, these authors relied on lateral movements along the surface of a touch-screen, meaning that participants pressed the touch-screen while conducting a lateral movement.”. 

References:

Werner, K., Raab, M., & Fischer, M. H. (2019). Moving arms: the effects of sensorimotor information on the problem-solving process. Thinking & Reasoning, 25(2), 171–191. https://doi.org/10.1080/13546783.2018.1494630

8. In the paragraph starting in line 667, the authors entertain a possible explanation of the absence of effects in the present study that strikes me as undermining the whole study: they say that perhaps the kind of force that they measured (pushing with two fingers in a given direction) may not be sensitive to high-level cognitive processes in the way that force in a precision grip has been already shown to be. If this is so, the authors are saying that they are using a measure that is invalid as an index of the cognitive processes of interest. They should strive to rule out this possibility, either with good arguments or by carrying out an experimental construct validation of the measure. If the authors finally accept that they cannot be sure that the measure is a valid one, then they should change their conclusions: the SNAs are absent either because there is no overt directional movement or because the measure is not valid.

Reply:

Thank you for your valuable feedback. We think that we have a valid measurement based on previous studies (Aravena et al., 2012; Aravena et al., 2014, Frak et al., 2010; Nazir et al., 2017) and recent evidence from the Miklashevsky et al. (2022) study that you suggested. In this study, it cannot be fully ruled out that this exact measure might have not been sensitive enough to pinpoint the SNAs in force data. However, we did show evidence against SNAs in our design in the RNG task with an independent measure: In all directions, no interaction between the small number bias and the direction of the force application was detected. Namely, our participants produced equally more small numbers in all directions, which is a classical signature of RNG. Due to the bidirectional nature of the embodiment perspective, such an interaction was to be expected in the case of reliable SNAs in our design. We have added an argument in lines 834-837 that now reads: “At the very least, isometric force served as a manipulation in any direction and ensured that participants deployed directional force [see limitations of Experiment 2 in 76]. Yet, in our independent assessment of H1, we do not find any SNAs in any direction during the RNG task.”.

Additionally, the discussion of Miklashevskys et al. (2022) paper (your point 6) should have addressed this reviewer’s concerns about the validity of the method and the implications for SNA research. 

In lines 318-320 we directly state that we utilize the same method as Miklashevsky et al. (2022). It now reads: ” Hereby, we utilized the same isometric force paradigm as Miklashevsky et al. [40]. Therefore, our isometric force paradigm does not require the execution of a lateralized response after processing a number.”.

References:

Aravena, P., Courson, M., Frak, V., Cheylus, A., Paulignan, Y., Deprez, V., & Nazir, T. A. (2014). Action relevance in linguistic context drives word-induced motor activity. Frontiers in Human Neuroscience, 8(1), 163. https://doi.org/10.3389/fnhum.2014.00163

Aravena, P., Delevoye-Turrell, Y., Deprez, V., Cheylus, A., Paulignan, Y., Frak, V., & Nazir, T. (2012). Grip Force Reveals the Context Sensitivity of Language-Induced Motor Activity during “Action Words” Processing: Evidence from Sentential Negation. PLoS ONE, 7(12), e50287. https://doi.org/10.1371/journal.pone.0050287

Frak, V., Nazir, T., Goyette, M., Cohen, H., & Jeannerod, M. (2010). Grip Force Is Part of the Semantic Representation of Manual Action Verbs. PLoS ONE, 5(3), e9728. https://doi.org/10.1371/journal.pone.0009728

Miklashevsky, A., Fischer, M. H., & Lindemann, O. (2022). Spatial-numerical associations without a motor response? Grip force says ‘Yes’. Acta Psychologica, 231, 103791. https://doi.org/10.1016/j.actpsy.2022.103791

Nazir, T. A., Hrycyk, L., Moreau, Q., Frak, V., Cheylus, A., Ott, L., Lindemann, O., Fischer, M. H., Paulignan, Y., & Delevoye-Turrell, Y. (2017). A simple technique to study embodied language processes: the grip force sensor. Behavior Research Methods, 49(1), 61–73. https://doi.org/10.3758/s13428-015-0696-7

Reviewer #2:

1. It seems that the central argument relies on the interpretation of a null effect. Of course, there are some instances where null effects are valuable, and the statistics were appropriate given the sort of conclusion the authors wished to make. Nevertheless, I had trouble making sense of this particular null effect, because I have no standard upon which to evaluate it. Are there cases where we *should* expect this method to reveal SNAs? If I'm understanding correctly, this approach has only been used successfully in the study of language. I'm not an expert on those studies or this methodology, but it seems that even the authors agree their approach is somewhat different from what is used in prior work. Therefore, I would be uncomfortable concluding that covert directional movements *cannot* induce SNAs if we do not have strong reason to believe that this design is sufficiently sensitive to detect covert directional movements to the relevant extent. In other words: I feel that we'd need a comparable positive effect to evaluate the lack of an effect here. Is there any way to demonstrate that this method *is* sensitive enough to detect some effect that we'd have strong reason to predict?

Reply:

Thank you for pointing this out. Indeed, we have argued that our measure is valid based on the grip force studies from linguistics. Now we have additional evidence for its validity based on a study published right after the submission of this manuscript, utilizing the same method (Miklashevsky et al., 2022). 

Miklashevsky et al. (2022) have utilized the same force measures as in the current study and demonstrated that it can be employed for studying embodied numerical processing. The authors found SNAs in their study, namely smaller numbers inducing more force in the left hand and larger numbers inducing more force in the right hand. This finding shows that SNAs can be found outside overt movements and with continuously employed isometric force. This has two major implications for our current study. First, the finding helps us extend the methodology to numerical cognition. Second, and at the same time, it challenges, as you rightfully point out, our conclusion that overt movements are necessary to induce SNAs. We have now integrated this new study in our revised Introduction and Discussion. This new study has enormously helped extending our argument and the limits of the current study and also the validity of our force measure. It also helped to change and adopt our conclusion on several occasions through the manuscript. The main changes in the Introduction are in lines 180-189 which now reads: 

“Very recently, a new study challenged the importance of active responses in the response space. Miklashevsky et al. [40] relied on a paradigm of passive and continuous isometric force readings during numerical tasks. Participants produced more force in the left hand while processing smaller numbers and more force in the right hand while processing larger numbers. These results suggest that active responses might not be required to induce SNAs. However, the authors did not control for spatial effects in their design. Specifically, in the response space, the authors utilized two sensors that were held in the left and in the right hand, thus being laterally displaced. Therefore, spatial information in the response space was apparently sufficient to elicit SNAs in their study. “

Further, in lines 318-320 we directly state that we utilize the same method as Miklashevsky et al. (2022) to show that the method is valid. It now reads: ” Hereby, we utilized the same isometric force paradigm as Miklashevsky et al. [40]. Therefore, our isometric force paradigm does not require the execution of a lateralized response after processing a number.”.

In the Discussion the changes are in lines 785-811 which now reads:

“Second, as stated in the Introduction, Miklashevsky et al. [40] found SNAs in numerical tasks during passive isometric force recordings utilizing the same method as in our study. The authors employed an experimental design utilizing two lateralized force sensors in a task that did not require active responses and instead recorded passive forces applied to the sensors. The authors found that smaller numbers induced more force in the left hand and larger numbers induced more force in the right hand. Such a finding is particularly intriguing because SNAs were produced without an active response and therefore not relying on an overt movement through space. This finding suggests that overt movements through space are not necessary in order to elicit SNAs. Instead, spatial information about effector positions by itself might be the sole contributor to SNAs formation [cf. 78]. Indeed, the authors utilized two spatially aligned sensors and did not control for spatial information in the response space.

Additionally, keep in mind that the authors employed two different tasks compared to our study. The authors themselves suggested that the task itself is critical in finding SNAs as they report weaker SNAs in their surface numerical decision task (“decide if this a number or a letter”) than in the classical magnitude classification task (“is this number larger or smaller than 5”). Therefore, it remains to be seen how their finding will translate to other experimental tasks such as random number generation and mental arithmetic as was the case in the current study. Together, Miklashevky et al.’s [40] study and our own can be interpreted as suggesting that active responses (and overt movements) might not be per se necessary to elicit SNAs, while spatial information might play a critical role. Future experiments could test more systematically if SNAs can exist in an experimental design space that excludes both, spatial information in the presentation and response space, as well as overt movements in the response space. Therefore, the current study contributes to the overarching discussion on design space and effector specification [79] necessary to elicit SNAs.”.

Additionally, we have further addressed your point by specifying isometric force in a different example in lines 771-779. Now it reads: “Indeed, as previously shown, an isometric force manipulation (but not measurement) during a lateral movement was successfully applied to induce spatial biases across the horizontal axis in a mathematical problem-solving task [Experiment 2 in 76]. Participants generated more addition solutions following a right-sided movement and more subtraction solutions following a left-sided movement. Notably, these authors relied on lateral movements along the surface of a touch-screen, meaning that participants pressed the touch-screen while conducting a lateral movement.”. This finding is also a positive finding that an isometric force manipulation (but not measurement) should be sufficient to induce SNAs. However, in our study this was not the case as our directional isometric force manipulation did not induce SNAs in the RNG task (see lines 668-676). We have further developed the argument in lines 833-836 that now reads: “At the very least, isometric force served as a manipulation in any direction and ensured that participants deployed directional force [see limitations of Experiment 2 in 76]. Yet, in our independent assessment of H1, we do not find any SNAs in any direction during the RNG task.”. 

Other adaptations of the conclusion:

1. Lines 31-33 “Here we argue that overt movements in space contribute to the formation of spatial-numerical associations (SNAs).”.

2. Lines 36-37 we have deleted the final sentence. It no longer reads: “We conclude that overt movements through space enable SNAs.”. 

3. Lines 291-293:” Generally, spatial information in the design (e.g., movements through space or responses in space) seems to be essential when investigating SNAs, both as a response producing SNAs and to activate SNAs.”.

4. Lines 300-302:” Considering all the presented evidence, we can conclude that spatial information is omnipresent in many experimental paradigms that study SNAs.”.

5. Lines 697-698: “Therefore, spatial information (e.g., movement through space) seems to be deeply rooted into SNAs assessment. To investigate the role of movement in space, future experiments could contrast designs utilizing movement in space against designs that utilize static and central versus static and lateral responses [for static and lateral responses see 40].”.

6. Lines 866-867: “We conclude that lateralized spatial information in the design space plays a crucial role in eliciting SNAs.”.

References:

Almaatouq, A., Griffiths, T. L., Suchow, J. W., Whiting, M. E., Evans, J., & Watts, D. J. (2022). Beyond Playing 20 Questions with Nature: Integrative Experiment Design in the Social and Behavioral Sciences. Behavioral and Brain Sciences, 1-55. https://doi.org/10.1017/S0140525X22002874

Werner, K., Raab, M., & Fischer, M. H. (2019). Moving arms: the effects of sensorimotor information on the problem-solving process. Thinking & Reasoning, 25(2), 171–191. https://doi.org/10.1080/13546783.2018.1494630

Miklashevsky, A., Fischer, M. H., & Lindemann, O. (2022). Spatial-numerical associations without a motor response? Grip force says ‘Yes’. Acta Psychologica, 231, 103791. https://doi.org/10.1016/j.actpsy.2022.103791

Rizzolatti, G., Riggio, L., & Sheliga, B. (1994). Space and Selective Attention. In Attention and Performance XV. https://doi.org/10.7551/mitpress/1478.003.0016

2. Is there something circular about the central argument? The authors argue that explicit movement is required to induce SNAs. But isn't it also possible that this measure is just not a measure of SNAs? In other words: Am I correct to understand that there is no independent evidence of SNAs in this task? If that's true, is it not possible that participants simply exhibited no SNAs, but that, if they had, this measure may have correctly detected them? In other words, there are two possible conclusions one could draw from the null effect. One conclusion is that subjects are not exhibiting SNAs. But another equally valid conclusion is that the subjects are exhibiting SNAs, but that this measure isn't sufficient to detect it. Perhaps, for instance, eye movements could have revealed SNAs even if these motor movements did not.

Reply:

Thank you for the valid comment and the concern. We have thoroughly addressed this issue in your point 1 and replying to the other reviewer. For clarification, there is independent evidence in the RNG task that no SNAs were activated in the study (see lines 668-676). Additionally, we have extended the argument. In lines 833-836 it now reads: “At the very least, isometric force served as a manipulation in any direction and ensured that participants deployed directional force [see limitations of Experiment 2 in 76]. Yet, in our independent measure of H1, we do not find any SNAs in any direction during the RNG task.”.

References:

Werner, K., Raab, M., & Fischer, M. H. (2019). Moving arms: the effects of sensorimotor information on the problem-solving process. Thinking & Reasoning, 25(2), 171–191. https://doi.org/10.1080/13546783.2018.1494630

3. In the arithmetic task, how are the equations appearing? The language in the Methods is not clear. It says the stimuli were presented "sequentially", but where? Were these stimuli arranged spatially in any way, or did they all appear in the same location one after the other? If the latter, what was the timing of that like? 

Ultimately, I think it could be valuable to have this result published. Perhaps others would be inspired to take a similar approach, which could prove valuable. However, I feel that the results should not be published without, at the very least, clearly qualifying the results. I'm just not sure that we could take from a single sort of null effect, in a very untested paradigm, that anything certainly is or certainly is not true. Then again, the authors may have some very strong arguments as to why they think these results do merit the conclusions that they made. Or it may be that I am misunderstanding something about the study. If either of these things were true, I would be able to more enthusiastically recommend publication.

Reply Part a:

Thank you for your suggestion. We have elaborated on the spatial information in the introduction in lines 344-347 that now reads: “However, we removed spatial information from the presentation and the response space. Therefore, instead of providing spatial information in the form of targets to probe these SNAs, we again relied on central stimuli presentation and covert directional force.”. Further, we have adjusted in the General Method section (lines 468-474) both, spatial and temporal information that now reads:” Participants were asked to watch mathematical equations (from here on, referred to as trials) in which individual stimuli (operand 1, operation sign, operand 2, equal sign, result) were presented sequentially after each other centrally on the computer screen. Some of the proposed results were wrong. In case of a wrong result participant had to reject the result by saying “No“. Each stimulus was shown for 500 ms, then the next stimulus appeared. There was an inter-trial interval of 1500 ms between the trials (Fig 3).”.

Reply Part b (“Ultimately…”):

Thank you for this comment. As you will read, we have implemented your comments in our manuscript and feel that it has now substantially improved. In your point 2 we have discussed why we think our method is valid in this current study and how our study contributes to the field of numerical cognition and the boundary conditions of SNAs.

On the general level, point 1 that you have raised is especially important in understanding how our results inform the field of numerical cognition. For this, we have implemented Miklashevsky et al. (2022) study in our Introduction and Discussion and contrasted their positive results to ours. We feel that with the integration of that study we found a strong argument on how to interpret boundary conditions of SNAs. We also suggest future directions on how to obtain more evidence on the relevance of overt movements and spatial information in the design space. On the general level, our manuscript and the findings strongly contribute to the overarching discussion on how to specify, rather than quantify psychological effects ensuring replicability and generalizability of experimental phenomena (Almaatouq et al., 2022). In lines 804-811 in now reads: “Together, Miklashevky et al.’s [40] study and our own can be interpreted as suggesting that active responses (and overt movements) might not be per se necessary to elicit SNAs, while spatial information might play a critical role. Future experiments could test more systematically if SNAs can exist in an experimental design space that excludes both, spatial information in the presentation and response space, as well as overt movements in the response space. Therefore, the current study contributes to the overarching discussion on design space and effector specification [79] necessary to elicit SNAs”.

References:

Almaatouq A, Griffiths TL, Suchow JW, Whiting ME, Evans J, Watts DJ. Beyond Playing 20 Questions with Nature: Integrative Experiment Design in the Social and Behavioral Sciences. Behav Brain Sci. 2022; 1–55. doi:10.1017/S0140525X22002874

Miklashevsky, A., Fischer, M. H., & Lindemann, O. (2022). Spatial-numerical associations without a motor response? Grip force says ‘Yes’. Acta Psychologica, 231, 103791. https://doi.org/10.1016/j.actpsy.2022.103791

Other:

1. I found it surprisingly difficult to understand the key manipulation. I spent much time wondering whether the experiment involved applying pressure to the subjects' hands, or whether the authors were measuring covert movements -- or, both. I'm still not sure. I think this stems from the ambiguity in phrases like "We studied continuous isometric forces..." It just wasn't clear to me whether the 'forces' referred to a force that was being applied by the device, or by the participant. I think this problem can be easily solved by just revisiting this language through the manuscript, to ensure that it would be clear to a reader that has no prior knowledge of the procedure or design.

Reply:

Thank you for the comment. We have written clarifying sentences in multiple sections.

1. Lines 312-318: “In the present experiment we centralized the presentation and the response space utilizing covert directional movement. We asked participants to push against a sensor surface, thus generating a covert movement. Here, we define a covert movement as an application of constant and continuous isometric force to a surface. Such a covert movement was exerted by the participant to a surface in a direction on the horizontal (left or right) and the vertical (up or down) axes during two tasks.”. 

2. Lines 344-347: “However, we removed spatial information from the presentation and the response space. Therefore, instead of providing spatial information in the form of targets to probe these SNAs, we again relied on central stimuli presentation and covert directional force.”. 

3. Lines 386-390: “A customized home-made wooden/metal box was built to enable covert directional movements (i.e., participants’ exerting isometric forces on a fixed force sensor surface) along the vertical and the horizontal axes (two levels each: left/right on a horizontal axis or up/down on the vertical axis).”.

2. I had trouble understanding what counts as 'overt' directional movement vs. 'covert' directional movement. In the classic SNARC design, is there overt directional movement? Subjects are just keeping their hands in one place. Why would this be overt? And here, if forces are being applied in one direction or the other, isn't that quite 'overt'? It's an explicit manipulation that the subjects would ostensibly be aware of. There are several other border cases that I'm just not sure about. For instance, there are a few papers from Stella Lourenco's lab that seem to fall somewhere in between:

Holmes, K. J., Ayzenberg, V., & Lourenco, S. F. (2016). Gamble on gaze: Eye movements reflect the numerical value of blackjack hands. Psychonomic Bulletin & Review, 23, 1974-1981.

Aulet, L. S., Yousif, S. R., & Lourenco, S. F. (2021). Spatial–numerical associations from a novel paradigm support the mental number line account. Quarterly Journal of Experimental Psychology, 74(10), 1829-1840.

Aulet, L. S., & Lourenco, S. F. (2018). The developing mental number line: Does its directionality relate to 5-to 7-year-old children’s mathematical abilities?. Frontiers in Psychology, 9, 1142.

In the first, eye movements alone were used as a measure of SNAs. In the second and third, placement errors in a spatial task were used as a measure of SNAs. The latter certainly required movement, but of a very different kind that in many SNARC paradigms. The former obviously involves a kind of movement, but surely the present study also involves eye movements -- so shouldn't SNAs be induced by that, regardless of the other manipulation?

I don't have any strong opinions about this, except that, having read the paper several times, I'm not sure what counts as covert vs. overt.

Reply:

Thank you for raising this question and introducing this interesting literature. Under an overt movement, we understand a movement through space that has a starting point and an endpoint. This also includes movements of response keys. A covert movement we define a movement that has no spatial component. We expanded the definition in the manuscript.

1. Overt movement: Lines 286-290: “Hereby, an overt movement is defined as a movement through space that has a starting point and an endpoint. For instance, an overt movement can be a rhythmic head movement to the left/right/up/down [e.g., 51] or a simple button press in a reaction time paradigm representing a location-related action (Fig 1, examples 1 to 4).“ 

2. Covert movement: Lines 312-318: “In the present experiment we centralized the presentation and the response space utilizing covert directional movement. We asked participants to push against a sensor surface, thus generating a covert movement. Here, we define a covert movement as an application of constant and continuous isometric force to a surface. Such a covert movement was exerted by the participant to a surface in a direction on the horizontal (left or right) and the vertical (up or down) axes during two tasks.”. 

With this definition, it is the case that the literature you suggest falls into the category of overt movements. As for Holmes et al. (2016), in our Fig 1 (examples 5 and 6, see also lines 174-179) we specifically address eye movements as overt movements. Also, eye movements should have played no role in our current study. For instance, the RNG task was performed with the eyes closed (lines 430-432). During the SDA task, stimuli were presented centrally and sequentially (lines 468-474). Therefore, eyes movements during the SDA task would have been task irrelevant (see also argument in the Discussion that only goal-directed behaviour might be detectable, lines 844-846). 

As for Aulet & Lourenco (2018) and Aulet et al. (2021), both designs have movements in the response space either by moving a mouse and clicking or moving the hand to touch a screen. Therefore, both would also fall into the category of an overt movement. 

3. Related to both points above: If the manipulation involves instructing participants to apply forces in different directions, how is that not overt? Sorry to be difficult about all of this; I find myself very confused.

Reply:

Thank you for the valid comment on the necessity to provide thorough definitions. We tried to make it more accessible and addressed it in response to your comment 2 above. 

Minor:

1. The first sentence of the abstract doesn't stand on its own. What does it mean that people "respond faster to the left", for instance? Respond to what?

Reply:

Thank you for pointing out this unclarity. We have paraphrased the sentence to „ People respond faster to smaller numbers in their left space and to larger numbers in their right space.” (Lines 30-31)

---

## [Decision Letter · Decision Letter 1]

2 May 2023

PONE-D-22-30799R1How to not induce SNAs: the insufficiency of covert directional movementPLOS ONE

Dear Dr. Kühne,

Thank you for submitting your manuscript to PLOS ONE. After careful consideration, we feel that it has merit but does not fully meet PLOS ONE’s publication criteria as it currently stands. Therefore, we invite you to submit a revised version of the manuscript that addresses the points raised during the review process.

We look forward to receiving your revised manuscript.

Kind regards,

Alessia Tessari, Ph.D.

Academic Editor

PLOS ONE

Journal Requirements:

Additional Editor Comments:

As you will read, the reviewers appreciated the revised version of your manuscript. However, one of the reviewers still asks for minor revisions.

Reviewers' comments:

Reviewer's Responses to Questions

**Comments to the Author**

1. If the authors have adequately addressed your comments raised in a previous round of review and you feel that this manuscript is now acceptable for publication, you may indicate that here to bypass the “Comments to the Author” section, enter your conflict of interest statement in the “Confidential to Editor” section, and submit your "Accept" recommendation.

Reviewer #1: (No Response)

Reviewer #2: (No Response)

2. Is the manuscript technically sound, and do the data support the conclusions?

Reviewer #1: Yes

Reviewer #2: Partly

3. Has the statistical analysis been performed appropriately and rigorously? 

Reviewer #1: Yes

Reviewer #2: Yes

4. Have the authors made all data underlying the findings in their manuscript fully available?

Reviewer #1: Yes

Reviewer #2: Yes

5. Is the manuscript presented in an intelligible fashion and written in standard English?

Reviewer #1: Yes

Reviewer #2: Yes

6. Review Comments to the Author

Reviewer #1: The manuscript has been improved in many ways. Still, I found a number of inadequacies and also have some suggestions for further improvement. I will mention them in the same order as they appear in the manuscript.

- Title: I am not convinced by the label “covert movement”. In my opinion, a movement is covert if it is planned and mentally simulated, but without it reaching the muscles. In this study the planned action does reach the muscles, and the participant does not plan to perform any movement, just a permanent pressure. It would be better to use a different label. “Isometric force” maybe a good alternative, although when I first read it I had difficulty to parse it. Maybe “static directional pressure” or some other option.

- line 54: “produces” is not adequate. Options: “supports”, “suggests”...

- lines 72-80 (polarity correspondence account): It would be important to cite the work by Santiago and Lakens (2015) that failed to support the polarity correspondence account. In general, I don’t think there is much point of describing and discussing all these theoretical approaches in the introduction of the paper, as the data are not going to be relevant to any of them and they are not taken up again in the discussion.

- line 84: not all embodied approaches presume that the relation between the abstract and the concrete domain must be bidirectional. Conceptual metaphor theory is an embodied theory and it suggests an asymmetrical relation. Moreover, not all abstract concepts are the same: some abstract concepts seem to be of a different kind to numbers, being more based on interoceptive experiences and language (see the recent review by Borghi, Shaki and Fischer, 2022).

- line 126: “become associated with space” does not seem to me to be the best way to put it. It is clear that numbers have an association with space, and this association is built because of the accummulation of experiences. I think what the authors mean is that they want to study the conditions under which the association between numbers and space is manifested in behavior.

- line 131: the authors claim that the “common ingredient” of studies showing SNAs is physical space. However, this is not always the case. For Shaki and Fischer (2018) the key ingredient is that either number or space is part of the explicit definition of the task.

- line 150: under these conditions, Shaki and Fischer (2018) DID find a vertical SNA in one experiment (but none in the other experiment).

- line 181: the authors try and explain the results by Miklashevsky by having spatial information in the response space. However, other studies such as Shaki and Fischer (2018) did not have any spatial information in the response space and found lateral SNAs in magnitud comparison and a vertical SNA in parity judgement in one out of two experiments. Moreover, there are a number of studies that just present a number and find SNAs in the pattern of eye movements over a blank screen. The authors should try and give a coherent explanation of the whole pattern of findings currently available.

- line 214: the cite to Pinto [24] does not seem to be correct, as it refers to the study with neglect patients. Actually, it would be important to include all the studies by Pinto (2019 a y b, and 2021) into the set of findings that the authors try and integrate in their review of the literature.

- line 221: the conclusion that follows from Pinto’s studies is not that left and right spatial codes are necessary for SNAs. Instead it is that both numbers and space must be present AND linked in the definition of the task.

- line 225-226: incorrect. Shaki and Fischer (2018) found absolutely no horizontal SNARC in the parity task.

- lines 228-230: not completely correct: the vertical SNARC in the parity task was found in one experiment but not in the other.

- line 293: it is not correct to say that spatial cueing induces vertical but not horizontal SNAs with centralized stimuli and responses (see the last two comments).

- line 306: it is asserted that the present study uses “the same isometric force paradigm as Miklashevsky”. Even though the same sensor is used, the fact that Miklashevsky used a two-handed precision grip is a potentially important difference in paradigm (as discussed by the authors at the end of the paper). Therefore, it is misleading to say here that the paradigm in the two studies was the same.

- line 395: this is two designs, not one.

- line 399-400: exchange H1 and H2.

- line 438: this is two designs, not one.

- lines 474-477: the time windows overlap.

- line 654: as I have pointed out above, “spatial information seems to be deeply rooted into SNAs assessment” fails to take into account Pinto’s studies.

- line 705: I can’t see how the argument in the prior lines (703-704) “fits the reasoning of “Absence of evidence is not evidence of absence”. Bayesian statistics are indeed able to quantify evidence for absence.

- line 738: again, the authors claim that Miklashevsky used the same method as in the present study. Although they also measured isometric force, the differences in methods are substantial.

- line 813-814: From the present study it does not follow that “lateralized spatial information in the design space plays a crucial role in eliciting SNAs”. This is because in the present design there is lateralized spatial information, as the participants are instructed to press the sensor in lateral (and vertical) directions, and because, in general, the study cannot conclude anything about what is essential for SNAs to arise. It can only conclude that directional static pressures are not able by themselves to make SNAs to arise.

Reviewer #2: The authors responded admirably to the feedback they received. I see no reason that the paper should not be accepted at this point.

Still, I want to comment on one issue. Previously, I'd found myself confused about the key distinction in the paper between 'overt' and 'covert' movement. The authors' reply about this point left me more confused, I think. Ordinarily I wouldn't care so much about this difference, but it is central to the paper. I'm not sure that (1) this distinction is so clear-cut as the authors imply, that (2) the distinction here is one that maps onto how most people would naturally think of the difference, nor that (3) this distinction is the relevant one. In addition, I felt that grouping all three of the studies I mentioned as 'overt' was very surprising to me. I genuinely wouldn't have expected that. For that reason, I was hoping the authors would have said slightly more in the paper about exactly what things are and are not overt, referring to more specific examples like these. (I found the information in Figure 1 to be valuable but hard to fully process, even with extensive knowledge of these studies; a non-expert might have even more trouble.)

I don't find this disqualifying, however, because the results can be understood and interpreted without respect to that distinction. For this reason, I only wish to suggest that the authors should once more reconsider whether this language is apt, or whether it is being explained in the best way. Maybe it is; I'm not sure. It stood out to me, so I wanted to make a note of it.

7. PLOS authors have the option to publish the peer review history of their article (what does this mean?). If published, this will include your full peer review and any attached files.

Reviewer #1: **Yes: **Julio Santiago

Reviewer #2: No

---

## [Author Response · Author response to Decision Letter 1]

17 May 2023

Reviewer #1: 

The manuscript has been improved in many ways. Still, I found a number of inadequacies and also have some suggestions for further improvement. I will mention them in the same order as they appear in the manuscript.

- Title: I am not convinced by the label “covert movement”. In my opinion, a movement is covert if it is planned and mentally simulated, but without it reaching the muscles. In this study the planned action does reach the muscles, and the participant does not plan to perform any movement, just a permanent pressure. It would be better to use a different label. “Isometric force” maybe a good alternative, although when I first read it I had difficulty to parse it. Maybe “static directional pressure” or some other option.

Reply:

Thanks to both reviewers for pointing this out. We have now completely removed the words “overt” and “covert” from the manuscript as both reviewers made similar points regarding the wording and how it led to confusion. We now use “isometric force” and “isometric directional force” and already applied them in the title. Additionally, we have reformulated the sentence with the aim of the study to make the idea more accessible before introducing “isometric directional force”. We have also contrasted movement through space with our isometric directional force manipulation early in the Discussion. 

We have renamed the manuscript to “How to not induce SNAs: the insufficiency of directional force”.

In lines 324-326, it now reads: 

“Will directional but non-spatial movement along the horizontal and vertical axes, thus pressing into a direction of left, right, up, or down, elicit SNAs?”.

In lines 328-338, we follow up with:

 “We asked participants to press against a sensor surface, thus recording their isometric force production. Such isometric force was exerted constantly and continuously on the sensor surface in a direction on the horizontal (left or right) and the vertical (up or down) axes during two tasks. Hereby, we utilized the same force sensors as Miklashevsky et al. [41] but altered the paradigm to record the production of isometric force by one-handed presses into a direction instead of grip force (from here on described as “isometric directional force”).” 

In lines 695-699, it now reads:

“Notable, our isometric directional force manipulation did not include a movement through space. Instead, isometric directional force utilized a subtler approach of directionally applied pressure attempting to elicit spatial codes.

- line 54: “produces” is not adequate. Options: “supports”, “suggests”...

Reply:

Corrected. In lines 54-56, it now reads:

“The reaction time paradigm is widely accepted and provides evidence for the association of smaller numbers with left space and larger numbers with right space [4].”

- lines 72-80 (polarity correspondence account): It would be important to cite the work by Santiago and Lakens (2015) that failed to support the polarity correspondence account. In general, I don’t think there is much point of describing and discussing all these theoretical approaches in the introduction of the paper, as the data are not going to be relevant to any of them and they are not taken up again in the discussion.

Reply:

We have referenced the Santiago and Lakens (2015) paper to address existing counterevidence for the polarity correspondence account. In lines 80-82, it now reads: 

“Therefore, SNARC would not be limited to the horizontal axis (but see [11] for empirical evidence against the polarity correspondence account).”

Regarding the Introduction. The theoretical approaches discussed in the Introduction seem important to keep allowing us to introduce the context in which the experiment is based on the existing literature. This approach reflects our planning in the methodology and facilitates the comprehension of our study for the readers. Thus, we tried to be concise in our Introduction but respectfully decline the request to eliminate our theoretical context. Moreover, to establish better theoretical coherence between Introduction and Discussion, we now briefly discuss that the theories may need to be extended to allow predictions in cases of an absent movement (see lines 725-735). 

References:

Santiago, J., & Lakens, D. (2015). Can conceptual congruency effects between number, time, and space be accounted for by polarity correspondence? Acta Psychologica, 156, 179–191. https://doi.org/10.1016/j.actpsy.2014.09.016

- line 84: not all embodied approaches presume that the relation between the abstract and the concrete domain must be bidirectional. Conceptual metaphor theory is an embodied theory and it suggests an asymmetrical relation. Moreover, not all abstract concepts are the same: some abstract concepts seem to be of a different kind to numbers, being more based on interoceptive experiences and language (see the recent review by Borghi, Shaki, and Fischer, 2022).

Reply:

Thank you for the suggestions. We now acknowledge the asymmetrical relation of numbers and space. In lines 102-105, it now reads:

“Notably, there is an asymmetrical relationship between space and numbers, in which the understanding of space is more fundamental. The meaning of numbers is often based on the experiences of space while this is not necessarily the case vice versa [12,13]. 

Additionally, we have integrated the review by Borghi, Shaki, and Fischer (2022) into an existing section where we compared the mechanisms behind the vertical and the horizontal SNAs (lines 249-268).

Concretely, in lines 261-265, it now reads:

“These linguistic practices that describe vertical SNAs differ in their contribution to the embodiment of numbers compared to the horizontal SNAs that seem to rely more on sensorimotor experiences such as finger counting. Moreover, situated factors like interoceptive signals play an additional role in the perception and production of numbers that also might contribute to SNAs ([50]; for review see [51]).

References: 

Belli, F., Felisatti, A., & Fischer, M. H. (2021). “BreaThink”: breathing affects production and perception of quantities. Experimental Brain Research, 239(8), 2489–2499. https://doi.org/10.1007/s00221-021-06147-z

Borghi, A. M., Shaki, S., & Fischer, M. H. (2023). Abstract concepts: external influences, internal constraints, and methodological issues. Psychological Research, 86(8), 2370–2388. https://doi.org/10.1007/s00426-022-01698-4

- line 126: “Become associated with space” does not seem to me to be the best way to put it. It is clear that numbers have an association with space, and this association is built because of the accummulation of experiences. I think what the authors mean is that they want to study the conditions under which the association between numbers and space is manifested in behavior.

Reply:

Corrected. In lines 133-135, it now reads:

“The goal of the present study is to test the boundary conditions under which the association of numbers and space is manifested in behavior.”.

- line 131: the authors claim that the “common ingredient” of studies showing SNAs is physical space. However, this is not always the case. For Shaki and Fischer (2018) the key ingredient is that either number or space is part of the explicit definition of the task.

Reply:

Thank you for pointing this out. We have now removed the term “physical space” and corrected it to “spatial information”. Spatial information is a broader term that covers the response space, presentation space, and cueing. 

In lines 140-145, it now reads:

“By this rationale, the common ingredient for most studies seems to be the activation of spatial information, namely under the experimental conditions of response space (lateral responses), presentation space (lateralized stimuli), or spatial cueing (arrows and/or instructions) during stimulus presentation (see Fig 1 for a visual overview of prototypical laboratory set-ups).” 

- line 150: under these conditions, Shaki and Fischer (2018) DID find a vertical SNA in one experiment (but none in the other experiment).

Reply:

This is correct; however, the reviewer seems to have mixed up the order of the references. Line 150 (162-163 in the revised marked manuscript) refers to the study by Pinto et al. (2021; reference number 30) where no SNAs were found in experiment 2. Reference number 31(lines 160-161) is for Shaki and Fischer (2018) only describing the experimental design with no description of the result. 

- line 181: the authors try and explain the results by Miklashevsky by having spatial information in the response space. However, other studies such as Shaki and Fischer (2018) did not have any spatial information in the response space and found lateral SNAs in magnitud comparison and a vertical SNA in parity judgement in one out of two experiments. Moreover, there are a number of studies that just present a number and find SNAs in the pattern of eye movements over a blank screen. The authors should try and give a coherent explanation of the whole pattern of findings currently available.

Reply:

Thank you for the comment. The studies that we refer to in the Introduction point to the inconsistency of the literature and how we use it to motivate our own research. For this, we chose the best-fitting examples. However, based on the latter comments of the reviewer, we have also extended the literature by integrating some new studies suggested by the reviewer (lines 226-248). Together, these studies motivate our research and we do not think that adding even more examples would benefit the paper. We think that from lines 244-246 we can already draw the necessary conclusion:

“Together, the above-presented studies point to the conclusion that both, number magnitude and spatial information must be coactivated to produce reliable SNAs [30,31,48,49].”. 

References to the additional studies included:

Pinto, M., Pellegrino, M., Marson, F., Lasaponara, S., Cestari, V., D’Onofrio, M., & Doricchi, F. (2021). How to trigger and keep stable directional Space–Number Associations (SNAs). Cortex, 134, 253–264. https://doi.org/10.1016/j.cortex.2020.10.020

Pinto, M., Pellegrino, M., Marson, F., Lasaponara, S., & Doricchi, F. (2019). Reconstructing the origins of the space-number association: spatial and number-magnitude codes must be used jointly to elicit spatially organised mental number lines. Cognition, 190, 143–156. https://doi.org/10.1016/j.cognition.2019.04.032

- line 214: the cite to Pinto [24] does not seem to be correct, as it refers to the study with neglect patients. Actually, it would be important to include all the studies by Pinto (2019 a y b, and 2021) into the set of findings that the authors try and integrate into their review of the literature.

Reply:

Thank you for pointing this out. We have verified the citation (Pinto et al. 2021) and can confirm its accuracy (reference number 30 in the revised manuscript). This was a single but important citation showing an experimental design of a centralized presentation and response space. Based on your suggestions and for the interested readers, we have now referenced other studies by Pinto that fulfill the same criteria and further strengthen our argument. In lines 233-237, it now reads:

“The authors’ interpretation was that the numbers will only spatially align across the mental number line when both, number magnitude is activated and spatial response codes are present (for experimental designs that also centralized the presentation and response space see [48,49]).”

References:

Pinto, M., Pellegrino, M., Lasaponara, S., Scozia, G., D’Onofrio, M., Raffa, G., Nigro, S., Arnaud, C. R., Tomaiuolo, F., & Doricchi, F. (2021). Number space is made by response space: Evidence from left spatial neglect. Neuropsychologia, 154. https://doi.org/10.1016/j.neuropsychologia.2021.107773

Pinto, M., Pellegrino, M., Marson, F., Lasaponara, S., Cestari, V., D’Onofrio, M., & Doricchi, F. (2021). How to trigger and keep stable directional Space–Number Associations (SNAs). Cortex, 134, 253–264. https://doi.org/10.1016/j.cortex.2020.10.020

Pinto, M., Pellegrino, M., Marson, F., Lasaponara, S., & Doricchi, F. (2019). Reconstructing the origins of the space-number association: spatial and number-magnitude codes must be used jointly to elicit spatially organised mental number lines. Cognition, 190, 143–156. https://doi.org/10.1016/j.cognition.2019.04.032

- line 221: the conclusion that follows from Pinto’s studies is not that left and right spatial codes are necessary for SNAs. Instead, it is that both numbers and space must be present AND linked in the definition of the task.

Reply:

Thank you for pointing this out. We have now added multiple sentences to incorporate this. 

In lines 233-237, it now reads:

“The authors’ interpretation was that the numbers will only spatially align across the mental number line when both, number magnitude is activated and spatial response codes are present (for experimental designs that also centralized the presentation and response space see [48,49]).”

In lines, 244-246:

“Together, the above-presented studies point to the conclusion that in which both, number magnitude and spatial information must be coactivated to produce reliable SNAs [30,31,48,49].”

- line 225-226: incorrect. Shaki and Fischer (2018) found absolutely no horizontal SNARC in the parity task.

Reply:

Corrected (line 241).

- lines 228-230: not completely correct: the vertical SNARC in the parity task was found in one experiment but not in the other.

Reply:

Thank you for the comment. We agree that our report was not complete as we do not present the full findings of the experiment because this goes beyond the scope of the paper. However, based on the reviewer’s other suggestions the section leads to the conclusion we needed to motivate our research. 

In lines 244-246, it now reads:

“Together, the above-presented studies point to the conclusion that both, number magnitude and spatial information must be coactivated to produce reliable SNAs [30,31,48,49].”.

- line 293: it is not correct to say that spatial cueing induces vertical but not horizontal SNAs with centralized stimuli and responses (see the last two comments).

Reply:

Corrected. In lines 319-321, it now reads: “Third, cognitive spatial cueing induces stronger vertical than horizontal SNAs during centralized stimulus presentation and responses [31].”

Additionally, we have corrected a sentence in lines 249-253 that now reads:

“One possible explanation for why vertical SNAs remained stronger than horizontal SNAs could be the hierarchical nature of body-related knowledge representations involved in the task, according to which different mechanisms induce different types of SNAs, referred to as grounded, embodied, or situated cognition [29].”.

- line 306: it is asserted that the present study uses “the same isometric force paradigm as Miklashevsky”. Even though the same sensor is used, the fact that Miklashevsky used a two-handed precision grip is a potentially important difference in paradigm (as discussed by the authors at the end of the paper). Therefore, it is misleading to say here that the paradigm in the two studies was the same.

Reply:

We appreciate your helpful suggestion and corrected the sentence. In lines 334-338, it now reads:

“Hereby, we utilized the same force sensors as Miklashevsky et al. [41] but altered the paradigm to record the production of isometric force by one-handed presses into a direction instead of grip force (from here on described as “isometric directional force”).“

- line 395: this is two designs, not one.

- line 399-400: exchange H1 and H2.

Reply to both points:

Thank you for pointing this out. In lines 441-450, now reads:

“In order to test the concept-motor interactions of H1 and H2 with the same task, we have utilized two within-subject designs. Both designs utilized one within-subject factor being the force direction (two levels each: left/right on a horizontal axis or up/down on the vertical axis). To test H1, number magnitude constituted the dependent variable with two magnitude levels: small and large. To test H2, number magnitude constituted the independent variable with two magnitude levels: small and large while continuous isometric force was the dependent variable.”.

Regarding the second point, we cannot exchange H1 and H2 as this would be incorrect. 

- line 438: this is two designs, not one.

In lines 490-494, we describe the design of the SDA task. This task does not test the bi-directionality of motor-concept interactions but only the effect of number magnitudes on force magnitudes. Therefore, it is one design. 

- lines 474-477: the time windows overlap.

Reply:

The time windows were constructed manually after the uninformative cluster permutation analysis of both tasks. They were constructed to capture mainly the effects in the time windows of 100-250 ms. However, we covered the duration until the end of the trial to explore the data in more detail. Mathematically, the time windows had to overlap during the last fragment of the SDA trials. In lines 480-487, we describe the full reasoning. Concretely, in lines 485-487 it reads:

 “These time-windows were constructed to capture mainly the effects in the time-windows of 100-250 ms that are associated with semantic activation after a critical stimulus [72].”. 

- line 654: as I have pointed out above, “spatial information seems to be deeply rooted into SNAs assessment” fails to take into account Pinto’s studies.

Reply:

Thank you for the comment. Earlier, we have acknowledged the position by Pinto et al. (2019, 2021a,2021b) in lines 244-246. However, in that specific paragraph we emphasize on the role of spatial information in the movement itself. 

We have rewritten the section to make the point clear. In lines 719-725, it now reads:

“Our finding is in contrast to other RNG studies that found SNAs along the horizontal and vertical axes by using either active movement ([59]; see also [56] for RNG across the horizontal axis; for review see [28]) or passive movement through space [60]. Therefore, it is likely that activating a spatial code in the response space, requires a movement with a spatial component, such as movement through space or movement allocated in space (e.g., contacting spatially aligned response buttons), to create SNAs.”.

- line 705: I can’t see how the argument in the prior lines (703-704) “fits the reasoning of “Absence of evidence is not evidence of absence”. Bayesian statistics are indeed able to quantify evidence for absence.

Reply:

Thank you for pointing this out. We have removed the sentence (lines 787-788). 

- line 738: again, the authors claim that Miklashevsky used the same method as in the present study. Although they also measured isometric force, the differences in methods are substantial.

Reply:

Thank you for pointing this out. We have removed the comparison to our study. Further differences from our study to Miklashevksy’s are discussed in the Limitations section. In lines 821-823, it now reads:

“Second, as stated in the introduction, Miklashevsky et al. [41] found SNAs in numerical tasks during bimanual and passive isometric force recordings.”. 

- line 813-814: From the present study it does not follow that “lateralized spatial information in the design space plays a crucial role in eliciting SNAs”. This is because in the present design there is lateralized spatial information, as the participants are instructed to press the sensor in lateral (and vertical) directions, and because, in general, the study cannot conclude anything about what is essential for SNAs to arise. It can only conclude that directional static pressures are not able by themselves to make SNAs arise.

Reply:

Thank you for pointing this out. We have now changed the concluding sentence in lines 900-901 that now reads as:

“We conclude that applying pressure into the direction of left, right, up, and down does not suffice in eliciting SNAs along the horizontal and the vertical axes.”.

Reviewer #2: 

The authors responded admirably to the feedback they received. I see no reason that the paper should not be accepted at this point.

Still, I want to comment on one issue. Previously, I'd found myself confused about the key distinction in the paper between 'overt' and 'covert' movement. The authors' reply about this point left me more confused, I think. Ordinarily I wouldn't care so much about this difference, but it is central to the paper. I'm not sure that (1) this distinction is so clear-cut as the authors imply, that (2) the distinction here is one that maps onto how most people would naturally think of the difference, nor that (3) this distinction is the relevant one. In addition, I felt that grouping all three of the studies I mentioned as 'overt' was very surprising to me. I genuinely wouldn't have expected that. For that reason, I was hoping the authors would have said slightly more in the paper about exactly what things are and are not overt, referring to more specific examples like these. (I found the information in Figure 1 to be valuable but hard to fully process, even with extensive knowledge of these studies; a non-expert might have even more trouble.)

I don't find this disqualifying, however, because the results can be understood and interpreted without respect to that distinction. For this reason, I only wish to suggest that the authors should once more reconsider whether this language is apt, or whether it is being explained in the best way. Maybe it is; I'm not sure. It stood out to me, so I wanted to make a note of it.

Reply:

Thanks to both reviewers for pointing this out. We have now completely removed the words “overt” and “covert” from the manuscript as both reviewers made similar points regarding the wording and how it led to confusion. We now use “isometric force” and “isometric directional force” and already applied it in the title. Additionally, we have reformulated the sentence with the aim of the study to make the idea more accessible before introducing” isometric directional force”. We have also contrasted movement through space with our isometric directional force manipulation early in the discussion. 

We have renamed the manuscript to “How to not induce SNAs: the insufficiency of directional force”.

In lines 324-326, it now reads: 

“Will directional but non-spatial movement along the horizontal and vertical axes, thus pressing into a direction of left, right, up, or down, elicit SNAs?”.

In lines 328-338, we follow up with:

 “We asked participants to press against a sensor surface, thus recording their isometric force production. Such isometric force was exerted constantly and continuously on the sensor surface in a direction on the horizontal (left or right) and the vertical (up or down) axes during two tasks. Hereby, we utilized the same force sensors as Miklashevsky et al. [41] but altered the paradigm to record the production of isometric force by one-handed presses into a direction instead of grip force (from here on described as “isometric directional force”). ”.

In lines 695-699, it now reads:

“Notable, our isometric directional force manipulation did not include a movement through space. Instead, isometric directional force utilized a subtler approach of directionally applied pressure attempting to elicit spatial codes.

Changes to the reference list

The reference list changed from the original submission to the current version. Beneath, the updated reference list is displayed. Red-coloured references are new additions while the orange references were removed from the reference list. The order has changed due to additions, references deletions, and the altered referencing order. 

Reference list changes from the first submission to the current version:

1. Dehaene S, Brannon EM. Space, time, and number: a Kantian research program. 2010. doi:10.1016/j.tics.2010.09.009

2. Dehaene S, Bossini S, Giraux P. The mental representation of parity and number magnitude. J Exp Psychol Gen. 1993;122: 371–396. doi:10.1037/0096-3445.122.3.371

3. Fischer MH, Shaki S. Spatial associations in numerical cognition-From single digits to arithmetic. Quarterly Journal of Experimental Psychology. Psychology Press Ltd; 2014. pp. 1461–1483. doi:10.1080/17470218.2014.927515

4. Wood GM de O, Willmes K, Nuerk H, Fischer MH. On the Cognitive Link between Space and Number: A Meta-Analysis of the SNARC Effect. Psychol Sci Q. 2008 [cited 19 Aug 2021]. Available: https://psycnet.apa.org/record/2009-00781-003

5. De Hevia MD, Veggiotti L, Streri A, Bonn CD. At Birth, Humans Associate “Few” with Left and “Many” with Right. Curr Biol. 2017;27: 3879-3884.e2. doi:10.1016/j.cub.2017.11.024

6. Rugani R, Lunghi M, Di Giorgio E, Regolin L, Dalla B, Vallortigara G, et al. A mental number line in human newborns. bioRxiv. 2017; 159335. doi:10.1101/159335

7. Felisatti A, Laubrock J, Shaki S, Fischer MH. A biological foundation for spatial–numerical associations: The brain’s asymmetric frequency tuning. Ann N Y Acad Sci. 2020;1477: 44–53. doi:10.1111/nyas.14418

8. Abrahamse E, Van Dijck JP, Fias W. How does working memory enable number-induced spatial biases? Front Psychol. 2016;7. doi:10.3389/fpsyg.2016.00977

9. Proctor RW, Cho YS. Polarity correspondence: A general principle for performance of speeded binary classification tasks. Psychol Bull. 2006;132: 416–442. doi:10.1037/0033-2909.132.3.416

10. Proctor RW, Xiong A. Polarity Correspondence as a General Compatibility Principle. Curr Dir Psychol Sci. 2015;24: 446–451. doi:10.1177/0963721415607305

11. Santiago J, Lakens D. Can conceptual congruency effects between number, time, and space be accounted for by polarity correspondence? Acta Psychol (Amst). 2015;156: 179–191. doi:10.1016/j.actpsy.2014.09.016

11.12. Lakoff G, Johnson M. Conceptual Metaphor in Everyday Language. J Philos. 1980;77: 453. doi:10.2307/2025464

12. 13. Lakoff G, Núñez RE. Where Mathematics Comes From How: the Embodied Mind Brings Mathematics into Being. New York: Basic Books; 2000. 

13. 14. Fischer MH. Finger counting habits modulate spatial-numerical associations. Cortex. 2008;44. doi:10.1016/j.cortex.2007.08.004

14. 15. Gunderson EA, Spaepen E, Gibson D, Goldin-Meadow S, Levine SC. Gesture as a window onto children’s number knowledge. Cognition. 2015;144: 14–28. doi:10.1016/j.cognition.2015.07.008

15. 16. Barrocas R, Roesch S, Gawrilow C, Moeller K. Putting a Finger on Numerical Development – Reviewing the Contributions of Kindergarten Finger Gnosis and Fine Motor Skills to Numerical Abilities. Front Psychol. 2020;11. doi:10.3389/fpsyg.2020.01012

16.17. Michirev A, Musculus L, Raab M. A Developmental Embodied Choice Perspective Explains the Development of Numerical Choices. Front Psychol. 2021;12: 3261. doi:10.3389/fpsyg.2021.694750

17. 18. Overmann KA. Finger-counting in the Upper Palaeolithic. Rock Art Res. 2021;31: 63–80. doi:10.31235/osf.io/wgbe5

18. 19. Fischer MH, Brugger P. When digits help digits: Spatial-numerical associations point to finger counting as prime example of embodied cognition. Front Psychol. 2011;2. doi:10.3389/fpsyg.2011.00260

19. 20. Hohol M, Wołoszyn K, Nuerk H-C, Cipora K. A large-scale survey on finger counting routines, their temporal stability and flexibility in educated adults. PeerJ. 2018;6: e5878. doi:10.7717/peerj.5878

20. 21. Wasner M, Moeller K, Fischer MH, Nuerk HC. Aspects of situated cognition in embodied numerosity: The case of finger counting. Cogn Process. 2014;15: 317–328. doi:10.1007/s10339-014-0599-z

21. 22. Cipora K, Soltanlou M, Reips UD, Nuerk HC. The SNARC and MARC effects measured online: Large-scale assessment methods in flexible cognitive effects. Behav Res Methods. 2019;51: 1676–1692. doi:10.3758/s13428-019-01213-5

22. 23. Sato M, Lalain M. On the relationship between handedness and hand-digit mapping in finger counting. Cortex. 2008;44: 393–399. doi:10.1016/j.cortex.2007.08.005

23. 24. Fabbri M, Guarini A. Finger counting habit and spatial-numerical association in children and adults. Conscious Cogn. 2016;40: 45–53. doi:10.1016/j.concog.2015.12.012

24. 25. Lindemann O, Alipour A, Fischer MH. Finger Counting Habits in Middle Eastern and Western Individuals: An Online Survey. J Cross Cult Psychol. 2011;42: 566–578. doi:10.1177/0022022111406254

25. 26. Lucidi A, Thevenot C. Do not count on me to imagine how I act: behavior contradicts questionnaire responses in the assessment of finger counting habits. Ann des Telecommun Telecommun. 2014;46: 1079–1087. doi:10.3758/s13428-014-0447-1

26. 27. Aleotti S, Di Girolamo F, Massaccesi S, Priftis K. Numbers around Descartes: A preregistered study on the three-dimensional SNARC effect. Cognition. 2020;195: 104111. doi:10.1016/j.cognition.2019.104111

27. 28. Winter B, Matlock T, Shaki S, Fischer MH. Mental number space in three dimensions. Neuroscience and Biobehavioral Reviews. Elsevier Ltd; 2015. pp. 209–219. doi:10.1016/j.neubiorev.2015.09.005

28. 29. Fischer MH. A hierarchical view of grounded, embodied, and situated numerical cognition. Cogn Process. 2012;13: 161–164. doi:10.1007/s10339-012-0477-5

29. 30. Pinto M, Pellegrino M, Lasaponara S, Scozia G, D’Onofrio M, Raffa G, et al. Number space is made by response space: Evidence from left spatial neglect. Neuropsychologia. 2021;154. doi:10.1016/j.neuropsychologia.2021.107773

30. 31. Shaki S, Fischer MH. Deconstructing spatial-numerical associations. Cognition. 2018;175: 109–113. doi:10.1016/j.cognition.2018.02.022

31. 32. Gevers W, Lammertyn J, Notebaert W, Verguts T, Fias W. Automatic response activation of implicit spatial information: Evidence from the SNARC effect. Acta Psychol (Amst). 2006;122: 221–233. doi:10.1016/j.actpsy.2005.11.004

32. 33. Ito Y, Hatta T. Spatial structure of quantitative representation of numbers: Evidence from the SNARC effect. Mem Cogn. 2004;32: 662–673. doi:10.3758/BF03195857

33. 34. Shaki S, Fischer MH. Multiple spatial mappings in numerical cognition. J Exp Psychol Hum Percept Perform. 2012;38: 804–809. doi:10.1037/a0027562

34. 35. Fischer MH. Cognitive representation of negative numbers. Psychol Sci. 2003;14: 278–282. doi:10.1111/1467-9280.03435

35. 36. Schwarz W, Keus IM. Moving the eyes along the mental number line: Comparing SNARC effects with saccadic and manual responses. Percept Psychophys. 2004;66: 651–664. doi:10.3758/BF03194909

36. 37. Hesse PN, Bremmer F. The SNARC effect in two dimensions: Evidence for a frontoparallel mental number plane. Vision Res. 2017;130: 85–96. doi:10.1016/j.visres.2016.10.007

37. 38. Liu Di, Cai D, Verguts T, Chen Q. The Time Course of Spatial Attention Shifts in Elementary Arithmetic. Sci Rep. 2017;7: 1–8. doi:10.1038/s41598-017-01037-3

38. 39. Myachykov A, Ellis R, Cangelosi A, Fischer MH. Ocular drift along the mental number line. Psychol Res. 2016;80: 379–388. doi:10.1007/s00426-015-0731-4

39. 40. Hartmann M, Mast FW, Fischer MH. Spatial biases during mental arithmetic: evidence from eye movements on a blank screen. Front Psychol. 2015;6: 12. doi:10.3389/fpsyg.2015.00012

41. Miklashevsky A, Fischer MH, Lindemann O. Spatial-numerical associations without a motor response? Grip force says ‘Yes.’ Acta Psychol (Amst). 2022;231: 103791. doi:10.1016/j.actpsy.2022.103791

42. Fischer MH, Castel AD, Dodd MD, Pratt J. Perceiving numbers causes spatial shifts of attention. Nat Neurosci. 2003;6: 555–556. doi:10.1038/nn1066

41. 43. McCrink K, Dehaene S, Dehaene-Lambertz G. Moving along the number line: Operational momentum in nonsymbolic arithmetic. Percept Psychophys. 2007;69: 1324–1333. doi:10.3758/BF03192949

42. 44. Fischer MH, Knops A. Attentional cueing in numerical cognition. Frontiers in Psychology. Frontiers Research Foundation; 2014. doi:10.3389/fpsyg.2014.01381

43. 45. Mathieu R, Epinat-Duclos J, Sigovan M, Breton A, Cheylus A, Fayol M, et al. What’s Behind a “+” Sign? Perceiving an Arithmetic Operator Recruits Brain Circuits for Spatial Orienting. Cereb Cortex. 2018;28: 1673–1684. doi:10.1093/cercor/bhx064

44. 46. Pinhas M, Shaki S, Fischer MH. Heed the signs: Operation signs have spatial associations. Q J Exp Psychol. 2014;67: 1527–1540. doi:10.1080/17470218.2014.892516

45. 47. Masson N, Letesson C, Pesenti M. Time course of overt attentional shifts in mental arithmetic: Evidence from gaze metrics. Q J Exp Psychol. 2018;71: 1009–1019. doi:10.1080/17470218.2017.1318931

48. Pinto M, Pellegrino M, Marson F, Lasaponara S, Doricchi F. Reconstructing the origins of the space-number association: spatial and number-magnitude codes must be used jointly to elicit spatially organised mental number lines. Cognition. 2019;190: 143–156. doi:10.1016/j.cognition.2019.04.032

49. Pinto M, Pellegrino M, Marson F, Lasaponara S, Cestari V, D’Onofrio M, et al. How to trigger and keep stable directional Space–Number Associations (SNAs). Cortex. 2021;134: 253–264. doi:10.1016/j.cortex.2020.10.020

50. Belli F, Felisatti A, Fischer MH. “BreaThink”: breathing affects production and perception of quantities. Exp Brain Res. 2021;239: 2489–2499. doi:10.1007/s00221-021-06147-z

51. Borghi AM, Shaki S, Fischer MH. Abstract concepts: external influences, internal constraints, and methodological issues. Psychol Res. 2022;86: 2370–2388. doi:10.1007/s00426-022-01698-4

52. Matheson HE, Barsalou LW. Embodiment and Grounding in Cognitive. 2018; 1–27. doi:10.1002/9781119170174.epcn310

47. 53. Raab M. Judgment, Decision-Making and Embodied Choices. Academic Publisher.; 2021. 

48. 54. Walsh V. A Theory of Magnitude: The parts that sum to number. The Oxford Handbook of Numerical Cognition. 2015. doi:1093/oxfordhb/9780199642342.013.64

49. 55. Walsh V. A theory of magnitude: Common cortical metrics of time, space and quantity. Trends Cogn Sci. 2003;7: 483–488. doi:10.1016/j.tics.2003.09.002

50. 56. Loetscher T, Schwarz U, Schubiger M, Brugger P. Head turns bias the brain’s internal random generator. Current Biology. Cell Press; 2008. pp. R60–R62. doi:10.1016/j.cub.2007.11.015

51. 57. Towse JN, Valentine JD. Random Generation of Numbers: A Search for Underlying Processes. Eur J Cogn Psychol. 1997;9: 381–400. doi:10.1080/713752566

52. 58. Shaki S, Fischer MH. Random walks on the mental number line. Exp Brain Res. 2014;232: 43–49. doi:10.1007/s00221-013-3718-7

53. 59. Winter B, Matlock T. More is up… and right: Random number generation along two axes. Proc Annu Meet Cogn Sci Soc. 2013. 

54. 60. Hartmann M, Grabherr L, Mast FW. Moving along the mental number line: Interactions between whole-body motion and numerical cognition. J Exp Psychol Hum Percept Perform. 2012;38: 1416–1427. doi:10.1037/a0026706

55. 61. Popper K. The Logic of Scientific Discovery. Basic Books; 1959. 

62. Büsch D, Hagemann N, Bender N. Das Lateral Preference Inventory: Itemhomogenität der deutschen Version. Zeitschrift für Sport. 2009;16: 17–28. doi:10.1026/1612-5010.16.1.17

56. 63. Mayr S, Erdfelder E, Buchner A, Faul F. A short tutorial of GPower. Tutor Quant Methods Psychol. 2007. Available: http://www.psycho.uni-duesseldorf.de/aap/projects/gpower.

57. 64. Brysbaert M. How many participants do we have to include in properly powered experiments? A tutorial of power analysis with reference tables. Journal of Cognition. Ubiquity Press; 2019. pp. 1–38. doi:10.5334/joc.72

58. 65. Frak V, Nazir T, Goyette M, Cohen H, Jeannerod M. Grip Force Is Part of the Semantic Representation of Manual Action Verbs. Gribble PL, editor. PLoS One. 2010;5: e9728. doi:10.1371/journal.pone.0009728

66. Nazir TA, Hrycyk L, Moreau Q, Frak V, Cheylus A, Ott L, et al. A simple technique to study embodied language processes: the grip force sensor. Behav Res Methods. 2017;49: 61–73. doi:10.3758/s13428-015-0696-7

59. 67. Krause F, Lindemann O. Expyriment: A Python library for cognitive and neuroscientific experiments. Behav Res Methods. 2014;46: 416–428. doi:10.3758/s13428-013-0390-6

60. 68. Vicario CM. Perceiving numbers affects the internal random movements generator. Sci World J. 2012;2012. doi:10.1100/2012/347068

61. 69. Di Bono MG, Zorzi M. The spatial representation of numerical and non-numerical ordered sequences: Insights from a random generation task. Q J Exp Psychol. 2013;66: 2348–2362. doi:10.1080/17470218.2013.779730

62. 70. Mathôt S, Schreij D, Theeuwes J. OpenSesame: An open-source, graphical experiment builder for the social sciences. Behavior Research Methods. Springer; 2012. pp. 314–324. doi:10.3758/s13428-011-0168-7

64. 71. Miklashevsky A. Catch the star! Spatial information activates the manual motor system. Myachykov A, editor. PLoS One. 2022;17: e0262510. doi:10.1371/journal.pone.0262510

65. 72. Pulvermüller F, Shtyrov Y, Hauk O. Understanding in an instant: Neurophysiological evidence for mechanistic language circuits in the brain. Brain and Language. 2009. pp. 81–94. doi:10.1016/j.bandl.2008.12.001

66. 73. Wagenmakers EJ, Marsman M, Jamil T, Ly A, Verhagen J, Love J, et al. Bayesian inference for psychology. Part I: Theoretical advantages and practical ramifications. Psychon Bull Rev. 2018;25: 35–57. doi:10.3758/s13423-017-1343-3

67. 74. van Ravenzwaaij D, Wagenmakers EJ. Advantages Masquerading as “Issues” in Bayesian Hypothesis Testing: A Commentary on Tendeiro and Kiers (2019). Psychol Methods. 2022;27: 451–465. doi:10.1037/met0000415

75. Loetscher T, Brugger P. Exploring number space by random digit generation. Exp Brain Res. 2007;180: 655–665. doi:10.1007/s00221-007-0889-0

68. 76. JASP Team. JASP (Version 0.16.4)[Computer software]. 2022. Available: https://jasp-stats.org/

70. 77. Johari K, Behroozmand R. Premotor neural correlates of predictive motor timing for speech production and hand movement: evidence for a temporal predictive code in the motor system. Exp Brain Res. 2017;235: 1439–1453. doi:10.1007/s00221-017-4900-0

71. 78. Engel AK, Maye A, Kurthen M, König P. Where’s the action? The pragmatic turn in cognitive science. Trends in Cognitive Sciences. Elsevier Current Trends; 2013. pp. 202–209. doi:10.1016/j.tics.2013.03.006

72. Rees M. On the future: Prospects for humanity. Princeton University Press; 2021. 

73. 79. Werner K, Raab M, Fischer MH. Moving arms: the effects of sensorimotor information on the problem-solving process. Think Reason. 2019;25: 171–191. doi:10.1080/13546783.2018.1494630

74. 80. Deeney C, O’Sullivan LW. Effects of cognitive loading and force on upper trapezius fatigue. Occup Med (Chic Ill). 2017;67: 678–683. doi:10.1093/occmed/kqx157

75. Aravena P, Courson M, Frak V, Cheylus A, Paulignan Y, Deprez V, et al. Action relevance in linguistic context drives word-induced motor activity. Front Hum Neurosci. 2014;8: 163. doi:10.3389/fnhum.2014.00163

76. Aravena P, Delevoye-Turrell Y, Deprez V, Cheylus A, Paulignan Y, Frak V, et al. Grip Force Reveals the Context Sensitivity of Language-Induced Motor Activity during “Action Words” Processing: Evidence from Sentential Negation. Paterson K, editor. PLoS One. 2012;7: e50287. doi:10.1371/journal.pone.0050287

77. 81. Rizzolatti G, Riggio L, Sheliga B. Space and Selective Attention. Attention and Performance XV. 1994. doi:10.7551/mitpress/1478.003.0016

82. Almaatouq A, Griffiths TL, Suchow JW, Whiting ME, Evans J, Watts DJ. Beyond Playing 20 Questions with Nature: Integrative Experiment Design in the Social and Behavioral Sciences. Behav Brain Sci. 2022; 1–55. doi:10.1017/S0140525X22002874

78. 83. Johansson RS. Sensory Control of Dexterous Manipulation in Humans. Hand and Brain. Elsevier; 1996. pp. 381–414. doi:10.1016/b978-012759440-8/50025-6

79. 84. Abolins V, Latash ML. Unintentional force drifts across the human fingers: implications for the neural control of finger tasks. Exp Brain Res. 2022;240: 751–761. doi:10.1007/s00221-021-06287-2

80. 85. Gibson JJ. The ecological approach to visual perception. Hilisdale, NJ: Lawrence Eribaum Associates; 1986. 

81. 86. Andres M, Davare M, Pesenti M, Olivier E, Seron X. Number magnitude and grip aperture interaction. Neuroreport. 2004;15: 2773–2777. 

82. 87. Lindemann O, Abolafia JM, Girardi G, Bekkering H. Getting a Grip on Numbers: Numerical Magnitude Priming in Object Grasping. J Exp Psychol Hum Percept Perform. 2007;33: 1400–1409. doi:10.1037/0096-1523.33.6.1400

83. 88. Moretto G, Di Pellegrino G. Grasping numbers. Exp Brain Res. 2008;188: 505–515. doi:10.1007/s00221-008-1386-9

84. 89. Hommel B, Müsseler J, Aschersleben G, Prinz W. The Theory of Event Coding (TEC): A framework for perception and action planning. Behav Brain Sci. 2001;24: 849–878. doi:10.1017/S0140525X01000103

---

## [Editor Report · Decision Letter 2]

19 Jun 2023

How to not induce SNAs: the insufficiency of directional force

PONE-D-22-30799R2

Dear Dr. Kühne,

We’re pleased to inform you that your manuscript has been judged scientifically suitable for publication and will be formally accepted for publication once it meets all outstanding technical requirements.

Kind regards,

Alessia Tessari, Ph.D.

Academic Editor

PLOS ONE
---

## [Editor Report · Acceptance letter]

21 Jun 2023

PONE-D-22-30799R2 

How to not induce SNAs: the insufficiency of directional force 

Dear Dr. Kühne:

I'm pleased to inform you that your manuscript has been deemed suitable for publication in PLOS ONE. Congratulations! Your manuscript is now with our production department. 

Kind regards, 

on behalf of

Professor Alessia Tessari 

Academic Editor

PLOS ONE